# 🔵BLoB: Bayesian Low-Rank Adaptation by Backpropagation for Large Language Models

**Yibin Wang** [*†1]   **Haizhou Shi** [*†1]   **Ligong Han** [12]   **Dimitris Metaxas** [1]   **Hao Wang** [†1]

## Abstract

Large Language Models (LLMs) often suffer from overconfidence during inference, particularly when adapted to downstream domain-specific tasks with limited data. Previous work addresses this issue by employing approximate Bayesian estimation *after* the LLMs are trained, enabling them to quantify uncertainty. However, such post-training approaches' performance is severely limited by the parameters learned *during* training. In this paper, we go beyond post-training Bayesianization and propose **B**ayesian **Lo**w-Rank Adaptation by **B**ackpropagation (**BLoB**), an algorithm that continuously and jointly adjusts both the mean and covariance of LLM parameters throughout the whole fine-tuning process. Our empirical results verify the effectiveness of BLoB in terms of generalization and uncertainty estimation, when evaluated on both in-distribution and out-of-distribution data. Code is available at `https://github.com/Wang-ML-Lab/bayesian-peft`.

## 1 Introduction

Despite the recent advancements in Large Language Models (LLMs) [9, 106, 105, 66, 15, 4, 91, 92, 77, 12, 1, 71], addressing the challenges of reliability and responsibility remains imperative [42, 100, 99]. LLMs often produce overconfident responses detached from factual grounding, posing potential harm to users [3, 107, 44, 43, 87, 50, 7, 118, 111, 120, 33, 68, 115, 45]. Therefore, accurately estimating response confidence (or uncertainty) is crucial to preemptively intervene before harm occurs. Current research predominantly focuses on eliciting the internal capability of uncertainty estimation of LLMs. For example, studies suggest that verbalized uncertainty yields better-calibrated results compared to conditional probability [87, 45].

While effective, the aforementioned methods do not offer a universal solution for expressing LLM uncertainty across all scenarios, especially when adapted [113] to domain-specific corpora, human preferences, or downstream tasks [44]. Even a well-calibrated LLM may struggle to estimate uncertainty during fine-tuning due to catastrophic forgetting of general knowledge [84]. Moreover, when applied to limited-scale downstream tasks, excessively over-parameterized LLMs can rapidly overfit, leading to overconfidence. Thus, enabling accurate uncertainty estimation of LLMs is vital for their reliable and responsible deployment.

Bayesian methods emerge as a natural solution for learning uncertainty estimation abilities among their counterparts [88, 11, 101, 98, 29, 46, 51, 61, 97, 58, 102, 20, 110]. These methods model predictive uncertainty $P(\boldsymbol{y}|\boldsymbol{x}, \mathcal{D})$ by marginalizing the posterior parameter distribution $P(\boldsymbol{\theta}|\mathcal{D})$ after observing the dataset $\mathcal{D}$:

$$P(\boldsymbol{y}|\boldsymbol{x}, \mathcal{D}) = \int P(\boldsymbol{y}|\boldsymbol{x}, \boldsymbol{\theta})P(\boldsymbol{\theta}|\mathcal{D})d\boldsymbol{\theta}. \tag{1}$$

However, adapting the Bayesian framework to LLMs poses significant challenges. LLM architectures typically incorporate complex components, including non-linear activation functions, rendering exact

---

*Equal Contribution. [1]Rutgers University. [2]MIT-IBM Watson AI Lab. †Correspondence to: Yibin Wang <yibin.wang@rutgers.edu>, Haizhou Shi <haizhou.shi@rutgers.edu>, Hao Wang <hw488@cs.rutgers.edu>.

38th Conference on Neural Information Processing Systems (NeurIPS 2024).

Bayesian inference of parameter posteriors intractable, i.e., unable to compute the integral precisely. Consequently, finding an accurate approximation algorithm for the true posterior distribution becomes a primary challenge. Additionally, modeling parameter posterior distributions demands extra memory space, imposing a prohibitive burden on systems due to the massive scale of LLMs.

Contemporary methods leverage Parameter-Efficient Fine-Tuning (PEFT) to reduce the number of tunable parameters, thus alleviating computational and storage resource burdens [23, 41, 26, 121, 56, 53]. Built on this, recent research explores Bayesianizing only the PEFT module during fine-tuning to calibrate LLMs [8, 103, 116, 69], somewhat relieving the burden of introducing more parameters for posterior approximation. However, initial investigations suggest that straightforward combinations of PEFT and basic Bayesian techniques like Monte-Carlo Dropout (MCD, [29]) or Deep Ensemble (ENS, [51, 8, 103]) yield only marginal improvements in generalization and uncertainty estimation. The most promising results to date involve Kronecker factorized Laplace approximation, applied after maximum a posteriori (MAP) estimation provided by any optimization algorithm [116]. Nevertheless, we argue that such post-training procedures bifurcate posterior approximation into two stages, inevitably leading to suboptimal estimation.

To address this challenge, we propose **B**ayesian **Lo**w-Rank Adaptation by **B**ackpropagation (**BLoB**), a Bayesian Deep Learning framework for fine-tuning LLMs with LoRA. BLoB jointly estimates the low-rank variational distributions' mean and covariance throughout the entire fine-tuning stage via backpropagation. Unlike methods relying on post-training approximation, BLoB enables simultaneous estimation of both the parameter mode (i.e., the mean if one assumes Gaussian distributions) and the parameter variance. Random sampling of model parameters based on variance estimation can enhance mode estimation. It thereby improves model performance in terms of accuracy and uncertainty estimation on both in-distribution and out-of-distribution datasets, as verified by our extensive experiments across multiple datasets. In summary, our contributions are:

- We propose a principled Bayesianization framework for Low-Rank Adaptation (LoRA) in Large Language Models (LLMs) by assuming that full weights' approximate posterior distribution has a low-rank structure containing a linear combination of independent Gaussian distributions.

- We show that, under mild conditions, optimization of the full-weight variational distribution can be done efficiently in the low-rank space of the weight update matrices.

- We introduce BLoB, a variational Bayesian low-rank adaptation framework for LLMs that jointly learns the mean and covariance of the variational distribution during fine-tuning.

- Extensive evaluations demonstrate the superiority of BLoB in terms of generalization and uncertainty estimation across different scenarios.

## 2 Preliminaries

In this section, we describe the notation as well as some preliminaries.

**Notation.** In this paper, scalars are denoted by lowercase letters, vectors by lowercase boldface letters, and matrices by uppercase boldface letters. Probability, expectation, and the dataset are denoted by $P$, $\mathbb{E}$, and $\mathcal{D}$, respectively. We use $[m] = \{1, 2, \cdots, m\}$ to denote the set of consecutive integer numbers starting from 1 and ending at $m$. For a matrix $\boldsymbol{X} = [\boldsymbol{x}_1, \cdots, \boldsymbol{x}_n] \in \mathbb{R}^{m \times n}$, we use $\text{vec}(\boldsymbol{X}) = [\boldsymbol{x}_1^\top, \boldsymbol{x}_2^\top, \cdots, \boldsymbol{x}_n^\top]^\top \in \mathbb{R}^{(mn) \times 1}$ to denote the vectorization operation; we use $\|\boldsymbol{X}\|_p = \left[\sum_{ij} |X_{ij}|^p\right]^{1/p}$ to define the $p$-norm of a matrix. We use $\otimes$ and $\circ$ to denote the Kronecker product and the element-wise product, respectively.

### 2.1 Low-Rank Adaptation (LoRA)

Inspired by the pioneering work on identifying and leveraging the low intrinsic rank of over-parameterized models during fine-tuning [55, 2], Low-Rank Adaptation (LoRA) assumes a low rank for the network's weight updates [41]. Typically in a single linear layer, LoRA decomposes each update matrix $\Delta \boldsymbol{W} = \boldsymbol{B}\boldsymbol{A}$ into the product of two low-rank matrices, where $\boldsymbol{B} \in \mathbb{R}^{m \times r}$ and $\boldsymbol{A} \in \mathbb{R}^{r \times n}$. Here, $m$, $n$, and $r$ denote the number of input neurons, output neurons, and the rank of the decomposition, respectively [41]. The forward pass of the linear layer with LoRA is formulated

as:

$$z = W_0 h + \Delta W h = W_0 h + BAh, \tag{2}$$

where $h$ and $z$ denote the input and output of the layer. Since the rank $r \ll \min\{m, n\}$ is significantly smaller than the numbers of input and output neurons (e.g., $r = 8 \ll m = n = 4096$ in the attention layer [41]), LoRA can drastically reduce the number of trainable parameters by approximately three orders of magnitude compared to full-parameter fine-tuning, while achieving comparable performance to the full-rank fine-tuning. This also leads to a similar reduction in memory consumption for storing optimizer states, thereby reducing the hardware requirements for fine-tuning LLMs to a great extent.

## 2.2 Variational Bayesian Networks (VBNs)

Bayesian Neural Networks (BNNs) estimate the posterior distributions of network parameters rather than relying on single-point estimates [10, 102]. Due to the intractability of exact inference of the true posterior, Variational Bayesian Networks (VBNs) approximate the true posterior using a variational distribution; this is done by minimizing its KL divergence from the true posterior distribution [39, 30, 11]. Specifically, if the weights $W$'s variational distribution $q(W|\theta)$ is parameterized by $\theta$, minimizing the divergence $\mathrm{KL}[q(W|\theta)\|P(W|\mathcal{D})]$ is equivalent to minimizing the following variational free energy with respect to $\theta$ [67, 117, 28]:

$$\mathcal{F}(\mathcal{D}, \theta) \triangleq -\mathbb{E}_{q(W|\theta)}[\log P(\mathcal{D}|W)] + \mathrm{KL}[q(W|\theta) \parallel P(W)]. \tag{3}$$

The final formulation of the objective function in Eqn. 3 offers another interpretation beyond minimizing the KL divergence between the variational and true posterior distributions [11]. Specifically, the first term maximizes the likelihood of the data, while the second term regularizes the variational distribution $q(W|\theta)$. We refer to the first term as the likelihood cost and the second term as the complexity cost. Optimizing these two terms involves balancing the expressiveness of the approximate posterior distribution and its simplicity.

Optimizing the first term of Eqn. 3 requires integrating out the parameterized variational distribution, necessitating Monte Carlo gradient estimation [52, 81]. Using this approach, we can incorporate the re-parameterization trick to enable backpropagation of the gradient to the underlying parameter $\theta$ [72, 47, 78]. In Bayes By Backprop (BBB) [11], the variational distribution is further simplified as a diagonal Gaussian $\mathcal{N}(\mu, \sigma^2)$, where $\sigma = \log(1 + \exp(\rho))$ ensures the standard deviation is positive. Then we have the Monte-Carlo estimation of Eqn. 3 that can pass the gradient to $\theta$:

$$\mathcal{F}(\mathcal{D}, \theta) \approx -\tfrac{1}{K} \sum_{k=1}^{K} \log P(\mathcal{D}|W_k) + \tfrac{1}{K} \sum_{k=1}^{K} [\log q(W_k|\theta) - \log P(W_k)], \tag{4}$$

where $W_k = \mu + \log(1 + \exp(\rho)) \odot \epsilon_k$ is the $k$-th sample of the weights yielded by parameterization and $\epsilon_k \sim \mathcal{N}(0, I)$. In BBB, the authors assume the prior distribution $P(W) = \pi \mathcal{N}(0, \sigma_1^2) + (1 - \pi)\mathcal{N}(0, \sigma_2^2)$ to be a mixture of Gaussians. Consequently, they optimize the second term based on weight sampling. In different scenarios, a simpler form of the prior, which allows for a closed-form solution, can also be considered. Although our proposed method is largely based on the existing framework of BBB, trivially combining BBB with LoRA does not yield satisfactory results. It is important to note that our specific designs are necessary to encourage the fast convergence of the variational distribution, which will be introduced later in Sec. 3.

## 3 Methodology

In this section, we formally introduce our proposed method, **B**ayesian **Lo**w-Rank Adaptation by **B**ackpropagation (BLoB). We begin by discussing the design choices for Bayesianizing LoRA parameters in Sec. 3.1, highlighting the assumptions BLoB makes about the approximate posterior in the full-weight space. Next, in Sec. 3.2, we explore the low-rank structure of the prior distribution in the full-weight space, which in turn motivates our choice of prior distributions in the low-rank parameter space. In Sec. 3.3, we introduce our parameterization method for the variational distributions. In Sec. 3.4, we integrate Flipout [108] into LoRA for improved sampling efficiency and faster convergence. Finally, we present the complete algorithmic description of BLoB in Sec. 3.5. Proof of the theorems and claims in this section can be found in Appendix A.

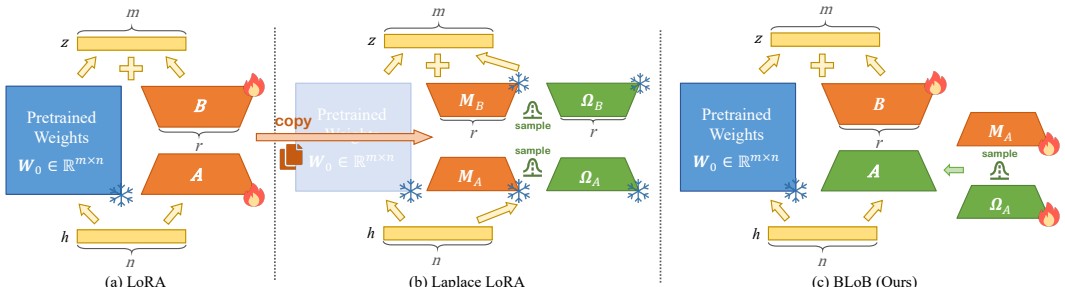

Figure 1: Overview of our Bayesian Low-Rank Adaptation by Backpropagation, i.e., BLoB **(right)** as well as comparison with existing methods such as LoRA **(left)** and Laplace LoRA **(middle)**.

### 3.1 Low-Rank Variational Approximate Posterior Distribution: LoRA Bayesianization

**Asymmetric LoRA Bayesianization.** In LoRA [41], the weights are treated asymmetrically. $A$ is randomly initialized, usually from the standard normal distribution or using Kaiming initialization, while $B = 0$ is initialized as a zero matrix to ensure that the model fully retains the capabilities of the pre-trained weights at the start of fine-tuning. The trivial solution of estimating the variational approximate posterior for the entire set of LoRA parameters can significantly hinder training convergence. For example, consider the Gaussian posteriors $q(A|\theta) = \mathcal{N}(A|M_A, \Omega_A^2)$ and $q(B|\theta) = \mathcal{N}(B|0, \Omega_B^2)$, where $\Omega_A$ and $\Omega_B$ are variance estimates added to $A$ and $B$, respectively. Although the expectation $\mathbb{E}_{A,B}[(W_0 + BA)x] = W_0 x + \mathbb{E}_{A,B}[BAx] = W_0 x$ preserves the functionality of the pre-trained model, accurate estimation requires an impractically large number of weight samples. Such variational distributions lead to significant fluctuations during the early stages of fine-tuning, unless the initial variance of $B$, $\Omega_B \to 0^+$, is intentionally minimized towards zero. Therefore, we take an *asymmetric* approach to initialize $\Omega_B = 0$ and keep it fixed throughout the fine-tuning process. This, in effect, gives up Bayesian modeling of the $B$ component and focuses only on the posterior of $A$ in LoRA, as shown in Fig. 1.

**Additional Advantages.** In addition to reducing sampling noise and improving convergence speed, our Bayesianization design has two further advantages. First, compared to modeling the variational distributions of both $A$ and $B$, our approach significantly reduces additional memory cost by approximately 50% per layer. Second, our design is equivalent to finding a posterior estimate for the full-weight matrix with a low-rank structure. For instance, by assuming a deterministic $B$ and Bayesianizing $P(A|\theta) = \mathcal{N}(A|M, \Omega^2)$, each element of the full weight matrix $W_{ij}$ is calculated as

$$W_{ij} = W_{0,ij} + \sum_{k=1}^{r} B_{ik} A_{kj}, \tag{5}$$

where $A_{kj} \sim \mathcal{N}(M_{kj}, \Omega_{kj}^2)$ is drawn independently $\forall k \in [r]$. It is noteworthy that due to the low-rank structure defined in Eqn. 5, the full-weight parameters of $W$ are no longer independent from each other. The correlation among them can be reflected by the following theorem:

**Theorem 3.1 (Variational Distribution of the Full-Weight Matrix in BLoB).** *With the pre-trained weight matrix $W_0 \in \mathbb{R}^{m \times n}$ and the low-rank weight update matrix $B \in \mathbb{R}^{m \times r}$, suppose that the variational distribution of the other low-rank update matrix $A \in \mathbb{R}^{r \times n}$ is Gaussian with $q(A|\theta = \{M, \Omega\}) = \prod_{ij} \mathcal{N}(A_{ij}|M_{ij}, \Omega_{ij}^2)$, where $M = [M_{ij}] \in \mathbb{R}^{r \times n}$ and $\Omega = [\Omega_{ij}] \in \mathbb{R}^{r \times n}$ are its mean and standard deviation, respectively. The equivalent variational distribution defined on the full weight matrix $W$ as in Eqn. 3 is given by*

$$q(\text{vec}(W)|B, \theta) = \mathcal{N}(\text{vec}(W)|\mu_q, \Sigma_q), \tag{6}$$

$$where \quad \mu_q = \text{vec}(W_0 + BM), \tag{7}$$

$$\Sigma_q = [I_n \otimes B] \cdot [\text{diag}(\text{vec}(\Omega)^2)] \cdot [I_n \otimes B^\top]. \tag{8}$$

Theorem 3.1 shows that our asymmetric LoRA Bayesianization is equivalent to using a Gaussian variational distribution for the full weight $W$ (i.e., Eqn. 6), with a flexible covariance matrix (i.e., Eqn. 8), to approximate the postetior distribution of the full weight $W$.

**Remark.** *The covariance matrix $\mathbf{\Sigma}_q$ is strictly singular, which consequently inspires us to design a prior $P(\mathbf{W})$ with such low-rank structure in Sec. 3.2. Previous work on low-rank Gaussians typically considers covariance with a similar structure $\mathbf{D}^2 + \mathbf{\Sigma}_q$, where $\mathbf{D}$ is diagonal [89, 70, 86, 90, 73]. However, sampling from a Gaussian with this structure requires sampling noise of the same shape as the full-weight matrix, which is not parameter-efficient; we therefore do not adopt this in our work.*

## 3.2 Low-Rank Prior Distribution

In Eqn. 3, optimizing the KL divergence between the variational and prior distributions in the space of full weights can be burdensome. Therefore, we assume the prior distribution of the full weights to be a low-rank Gaussian, with its mean centered at the pre-trained weights $\text{vec}(\mathbf{W}_0)$ and its covariance matrix parameterized by a rank-$r'$ matrix $\widetilde{\mathbf{R}} \in \mathbb{R}^{(mn) \times r'}$:

$$P(\text{vec}(\mathbf{W})) = \mathcal{N}(\text{vec}(\mathbf{W})|\boldsymbol{\mu}_p, \mathbf{\Sigma}_p),$$
$$\text{where} \quad \boldsymbol{\mu}_p = \text{vec}(\mathbf{W}_0), \tag{9}$$
$$\mathbf{\Sigma}_p = \widetilde{\mathbf{R}}\widetilde{\mathbf{R}}^\top.$$

Assuming a low-rank prior distribution and designing an appropriate $\widetilde{\mathbf{R}}$ allows us to optimize the KL divergence in the decomposed low-rank weight space, as suggested by the following theorem.

**Theorem 3.2** (**Efficient Computation of Full-Weight KL Divergence**). *Suppose the pre-trained weights $\mathbf{W}_0$, update matrix $\mathbf{B}$, and the variational distribution $q(\mathbf{A}|\boldsymbol{\theta})$ are defined as in Theorem 3.1, and the prior distribution of the full-weight matrix $P(\text{vec}(\mathbf{W}))$ is defined as Eqn. 9. Consider the Gaussian prior distribution $P(\mathbf{A}) = \prod_{ij} \mathcal{N}(A_{ij}|0, \sigma_p^2)$; we then have:*

$$\text{KL}[q(\text{vec}(\mathbf{W})|\mathbf{B}, \boldsymbol{\theta})\|P(\text{vec}(\mathbf{W}))] = \text{KL}[q(\mathbf{A}|\boldsymbol{\theta})\|P(\mathbf{A})], \tag{10}$$

*if $\widetilde{\mathbf{R}} = [\sigma_p \mathbf{I}_n \otimes \mathbf{R}]$, where $\mathbf{R}$ satisfies $\mathbf{R}\mathbf{R}^\top = \mathbf{B}\mathbf{B}^\top$.*

Theorem 3.2 shows that with a proper $\widetilde{\mathbf{R}}$, one can compute the KL divergence for the high-dimensional full weight $\text{vec}(\mathbf{W})$ simply by computing the KL divergence for $\mathbf{A}$, which is much lower-dimension, more parameter-efficient, more memory-efficient, and faster. Note that the Gaussian distributions we define for both the prior and the posterior are degenerate. However, they are valid for probabilistic inference [83], as (i) their probability density is well-defined, and (ii) their KL divergence is computable under the assumptions of Theorem 3.2. See Appendix A.1 for a detailed discussion.

Concretely, we assume that the prior distribution in BLoB follows the low-rank structure described in Theorem 3.2 and minimize the KL divergence term for the low-rank component $\mathbf{A}$ using its analytical solution in Eqn. 3:

$$\text{KL}[q(\mathbf{A}|\boldsymbol{\theta} = \{\mathbf{M}, \mathbf{\Omega}\})\|P(\mathbf{A})] = \frac{1}{2\sigma_p^2}(\|\mathbf{M}\|_2^2 + \|\mathbf{\Omega}\|_2^2) - \sum_{ij} \log \Omega_{ij}. \tag{11}$$

## 3.3 Parameterization of the Low-Rank Variational Distribution

The parameterization of the Gaussian variational distribution $q(\mathbf{A}|\boldsymbol{\theta})$ significantly affects the convergence speed of the KL term in Eqn. 11. The mean matrix $\mathbf{M}$ of $q(\mathbf{A}|\boldsymbol{\theta})$ has no additional constraints, we therefore parameterize it directly as the output of a neural network. Each entry of $q(\mathbf{A}|\boldsymbol{\theta})$'s diagonal covariance matrix $\mathbf{\Omega}$ (i.e., standard deviation) is non-negative; we therefore use element-wise parameterization $\Omega_{ij} = G_{ij}^2$, where $\mathbf{G} = [G_{ij}] \in \mathbb{R}^{r \times n}$ is the real parameter matrix that determines the standard deviation $\mathbf{\Omega}$. Since $\mathbf{\Omega}$ is usually initialized with small positive values close to zero, our parameterization method provides large gradients initially, contributing to the rapid decrease of the KL term. We further show, both theoretically and empirically, that our parameterization method, unlike BBB's softplus function $\log(1 + \exp(\cdot))$, is crucial for the fast convergence of $\mathbf{\Omega}$ when $q(\mathbf{A}|\boldsymbol{\theta})$ is close to the prior distribution $P(\mathbf{A})$ (see more analysis in Appendix A.2).

## 3.4 On Improving the Sample Efficiency of BLoB

**Improving Sample Efficiency with Flipout.** One main challenge in estimating the variational distribution (i.e., the approximate posterior) during fine-tuning lies in the sample efficiency of the

---

**Algorithm 1  Bayesian Low-Rank Adaptation by Backpropagation (BLoB)**

---

**Require:** dataset $\mathcal{D}$, pre-trained weight $\boldsymbol{W}_0$, low-rank component $\boldsymbol{B}$, $\boldsymbol{\theta} = \{\boldsymbol{M}, \boldsymbol{G}\}$ for parameterizing the mean and variance of $\boldsymbol{A}$;
**Require:** prior standard deviation $\sigma_p$, initialization hyperparameter $\epsilon$, number of input features $n$;
**Require:** number of samples during training $K$, number of iterations $T$, learning rate $\eta$;

1: $\boldsymbol{G} \sim \mathcal{U}(\frac{\epsilon}{\sqrt{2}}, \epsilon)$, $\boldsymbol{M} \sim \mathcal{U}\left(-\sqrt{\frac{6}{n}}, \sqrt{\frac{6}{n}}\right)$       ▷ Initialization of $\boldsymbol{A}$'s parameters.
2: $\boldsymbol{B} \leftarrow \boldsymbol{0}$       ▷ Initialization of $\boldsymbol{B}$.
3: **for** $t = 1, \cdots, T$ **do**
4:     Sample a mini-batch of data $\mathcal{D}_t \sim \mathcal{D}$ containing $b$ samples.
5:     **for** $k = 1, \cdots, K$ **do**
6:         Sample batched noise $\boldsymbol{E}_k \sim \mathcal{N}(\boldsymbol{0}, \boldsymbol{I})$.       ▷ Sample the noise.
7:         Let $\{\widetilde{\boldsymbol{E}}_{kj}\}_{j=1}^b \leftarrow \text{BLoBFlipout}(\boldsymbol{E}_k)$.       ▷ Eqn. 12
8:         Let $\{\boldsymbol{A}_{kj} = \boldsymbol{M} + \boldsymbol{G}^2 \circ \widetilde{\boldsymbol{E}}_{kj}\}_{j=1}^b$.
9:     **end for**
10:    Let $\widehat{\mathcal{F}}_t = -\frac{1}{Kb} \sum_{k=1}^K \sum_{j=1}^b \log P(\mathcal{D}_t | \boldsymbol{A}_{kj}, \boldsymbol{B}) + \frac{1}{2\sigma_p^2}(\|\boldsymbol{M}\|_2^2 + \|\boldsymbol{G}\|_2^4) - 2\sum_{ij} \log G_{ij}$.
11:                                                           ▷ Eqn. 13 and 11.
12:    Calculate the gradient w.r.t. the parameters:
        $\Delta_{\boldsymbol{M}} = \partial\widehat{\mathcal{F}}_t/\partial\boldsymbol{M}$, $\Delta_{\boldsymbol{G}} = \partial\widehat{\mathcal{F}}_t/\partial\boldsymbol{G}$, $\Delta_{\boldsymbol{B}} = \partial\widehat{\mathcal{F}}_t/\partial\boldsymbol{B}$.
13:    Update the parameters:
        $\boldsymbol{M} \leftarrow \boldsymbol{M} - \eta\Delta_{\boldsymbol{M}}$;
        $\boldsymbol{G} \leftarrow \boldsymbol{G} - \eta\Delta_{\boldsymbol{G}}$;
        $\boldsymbol{B} \leftarrow \boldsymbol{B} - \eta\Delta_{\boldsymbol{B}}$.
14: **end for**

---

weights [109, 25, 58]. During mini-batch stochastic gradient descent, a batch of examples typically share the same weights drawn from the variational distribution. This can lead to slow convergence of the likelihood cost in Eqn. 3. Drawing inspiration from [108], we introduce the technique of flipout to speed up the sampling procedure of our low-rank variational distributions $q(\boldsymbol{A}|\boldsymbol{\theta})$.

**LoRA Flipout.** Unlike the original approach, which applies rank-1 random flipping to the full weights, we apply flipout exclusively to the low-rank component $\boldsymbol{A}$. Specifically, suppose we have a mini-batch of input vectors $\boldsymbol{H} \in \mathbb{R}^{n \times b}$, where $b$ represents the batch size. We randomly sample two low-rank flipping matrices $\boldsymbol{S} \in \{-1, +1\}^{n \times b}$ and $\boldsymbol{T} \in \{-1, +1\}^{b \times r}$. Denoting as $\boldsymbol{E} \in \mathbb{R}^{r \times n}$ the weight noise sampled for this mini-batch, the batched output $\boldsymbol{Z}$ after applying flipout is then

$$\boldsymbol{Z} = \boldsymbol{W}_0\boldsymbol{H} + \boldsymbol{B}(\boldsymbol{M}\boldsymbol{H} + [(\boldsymbol{E} \circ \boldsymbol{\Omega})(\boldsymbol{H} \circ \boldsymbol{S})] \circ \boldsymbol{T}), \tag{12}$$

It is crucial that the independent noises added to the low-rank weight noise $\Delta\boldsymbol{A} \triangleq \boldsymbol{E} \circ \boldsymbol{\Omega}$ ensure *sampling independence across examples* within a mini-batch, thereby enhancing the sampling efficiency of the algorithm. This is done without violating the assumptions outlined in Theorem 3.1 and 3.2. As illustrated in algorithm 1, we use $\widetilde{\boldsymbol{E}}_{kj}$ to represent the equivalent noise applied to parameter $\boldsymbol{A}$ for the $j$-th example in the $k$-th batch after BLoBFlipout. Due to the low-rank structure of our Bayesianization method, the computational overhead of employing flipout in BLoB is also minimal.

### 3.5  BLoB: Final Algorithm

We are now ready to present our full BLoB algorithm.

**During training**, under the assumptions outlined in Theorem 3.1 and 3.2, optimizing the evidence lower bound on the full weight $\boldsymbol{W}$ can be efficiently done in the low-rank space, using the following final objective function:

$$\begin{aligned}
\mathcal{F}(\mathcal{D}, \boldsymbol{B}, \boldsymbol{\theta}) &= -\mathbb{E}_{q(\boldsymbol{W}|\boldsymbol{B},\boldsymbol{\theta})}[\log P(\mathcal{D}|\boldsymbol{W})] + \text{KL}[q(\boldsymbol{W}|\boldsymbol{B}, \boldsymbol{\theta}) \parallel P(\boldsymbol{W})] \\
&= -\mathbb{E}_{q(\boldsymbol{A}|\boldsymbol{\theta})}[\log P(\mathcal{D}|\boldsymbol{A}, \boldsymbol{B})] + \text{KL}[q(\boldsymbol{A}|\boldsymbol{\theta}) \parallel P(\boldsymbol{A})],
\end{aligned} \tag{13}$$

where $\boldsymbol{\theta} = \{\boldsymbol{M}, \boldsymbol{\Omega}\}$ denotes the set of the parameters underlying the variational distribution of the low-rank matrix $\boldsymbol{A}$. Additionally, to trade off between data fitting and posterior approximation,

Table 1: **Performance of different methods applied to LoRA on Llama2-7B pre-trained weights,** where Accuracy (**ACC**) and Expected Calibration Error (**ECE**) are reported in percentages. The evaluation is done across six common-sense reasoning tasks with a shared hyper-parameter setting after 5,000 gradient steps. We use $N$ to represent the number of samples during inference in BLoB. "↑" and "↓" indicate that higher and lower values are preferred, respectively. **Boldface** and underlining denote the best and the second-best performance, respectively.

| Metric | Method | Datasets | | | | | |
|---|---|---|---|---|---|---|---|
| | | WG-S [82] | ARC-C [18] | ARC-E [18] | WG-M [82] | OBQA [65] | BoolQ [17] |
| ACC (↑) | MLE | 68.99±0.58 | 69.10±2.84 | 85.65±0.92 | 74.53±0.66 | 81.52±0.25 | 86.53±0.28 |
| | MAP | 68.62±0.71 | 67.59±0.40 | 86.55±0.55 | 75.61±0.71 | 81.38±0.65 | 86.50±0.41 |
| | MCD [29] | 69.46±0.62 | 68.69±1.30 | 86.21±0.46 | **76.45±0.04** | 81.72±0.10 | 87.29±0.13 |
| | ENS [51, 8, 103] | 69.57±0.66 | 66.20±2.01 | 84.40±0.81 | 75.32±0.21 | 81.38±0.91 | 87.09±0.11 |
| | BBB [11] | 56.54±7.87 | 68.13±1.27 | 85.86±0.74 | 73.63±2.44 | 82.06±0.59 | **87.21±0.22** |
| | LAP [116] | 69.20±1.50 | 66.78±0.69[1] | 80.05±0.22 | 75.55±0.36 | 82.12±0.67 | 86.95±0.09 |
| | BLoB (N=0) | **70.89±0.82** | **70.83±1.57** | **86.68±0.60** | 74.55±1.94 | **82.73±0.41** | 86.80±0.23 |
| | BLoB (N=5) | 66.30±0.62 | 67.34±1.15 | 84.74±0.33 | 72.89±1.25 | 81.79±0.94 | 86.47±0.15 |
| | BLoB (N=10) | 69.07±0.34 | 68.81±1.09 | 85.56±0.35 | 73.69±0.17 | 81.52±0.74 | 86.99±0.24 |
| ECE (↓) | MLE | 29.83±0.58 | 29.00±1.97 | 13.12±1.39 | 20.62±0.74 | 12.55±0.46 | 3.18±0.09 |
| | MAP | 29.76±0.87 | 29.42±0.68 | 12.07±0.55 | 23.07±0.14 | 13.26±0.82 | 3.16±0.23 |
| | MCD [29] | 27.98±0.44 | 27.53±0.80 | 12.20±0.56 | 19.55±0.47 | 13.10±0.11 | 3.46±0.16 |
| | ENS [51, 8, 103] | 28.52±0.55 | 29.16±2.37 | 12.57±0.58 | 20.86±0.43 | 15.34±0.27 | 9.61±0.24 |
| | BBB [11] | 21.81±12.95 | 26.23±1.47 | 12.28±0.58 | 15.76±4.71 | 11.38±1.07 | 3.74±0.10 |
| | LAP [116] | **4.15±1.12** | 16.25±2.61[1] | 33.29±0.57 | 7.40±0.27 | 8.70±1.77 | **1.30±0.33** |
| | BLoB (N=0) | 20.62±0.83 | 20.61±1.16 | 9.43±0.38 | 11.23±0.69 | 8.36±0.38 | 2.46±0.07 |
| | BLoB (N=5) | 10.89±0.83 | 11.22±0.35 | 6.16±0.23 | 4.51±0.35 | **3.40±0.57** | 1.63±0.35 |
| | BLoB (N=10) | 9.35±1.37 | **9.59±1.88** | **3.64±0.53** | **3.01±0.12** | 3.77±1.47 | 1.41±0.19 |
| NLL (↓) | MLE | 3.17±0.37 | 2.85±0.27 | 1.17±0.13 | 0.95±0.07 | 0.73±0.03 | 0.32±0.00 |
| | MAP | 2.46±0.34 | 2.66±0.11 | 0.90±0.05 | 1.62±0.29 | 0.75±0.01 | 0.33±0.00 |
| | MCD [29] | 2.79±0.53 | 2.67±0.15 | 1.00±0.14 | 1.02±0.03 | 0.77±0.03 | 0.31±0.00 |
| | ENS [51, 8, 103] | 2.71±0.08 | 2.46±0.22 | 0.82±0.03 | 1.25±0.03 | 1.06±0.04 | 0.57±0.02 |
| | BBB [11] | 1.40±0.55 | 2.23±0.04 | 0.91±0.06 | 0.84±0.15 | 0.66±0.05 | **0.31±0.00** |
| | LAP [116] | **0.60±0.00** | 1.03±0.04[1] | 0.88±0.00 | 0.57±0.00 | 0.52±0.01 | **0.31±0.00** |
| | BLoB (N=0) | 0.91±0.10 | 1.19±0.02 | 0.56±0.01 | 0.60±0.01 | 0.56±0.02 | 0.32±0.00 |
| | BLoB (N=5) | 0.68±0.01 | 0.90±0.01 | 0.46±0.02 | 0.56±0.01 | 0.53±0.01 | 0.32±0.00 |
| | BLoB (N=10) | 0.63±0.01 | **0.78±0.02** | **0.40±0.01** | **0.54±0.00** | 0.50±0.01 | 0.31±0.00 |

we employ a KL re-weighting scheme, which is detailed in Appendix B.1. The full algorithmic description of BLoB training is shown in Algorithm 1.

**During inference,** for an input $\boldsymbol{x}$, we approximate the expected output distribution $P(\boldsymbol{y}|\boldsymbol{x})$ of BLoB by drawing $N$ samples from the variational distribution $q(\boldsymbol{W}|\boldsymbol{\theta})$. Empirically, $N = 10$ provides a good balance between estimation quality and computational efficiency:

$$\mathbb{E}_{q(\boldsymbol{W}|\boldsymbol{\theta})}[P(\boldsymbol{y}|\boldsymbol{x}, \boldsymbol{W})] \approx \tfrac{1}{N} \sum_{n=1}^{N} P(\boldsymbol{y}|\boldsymbol{x}, \boldsymbol{W}_n), \quad \boldsymbol{W}_n \sim q(\boldsymbol{W}|\boldsymbol{\theta}). \tag{14}$$

## 4 Experiments

In this section, we compare our BLoB with existing methods on real-world datasets. Sec. 4.1 introduces the experimental settings, including baselines, fine-tuning, and evaluation protocols. We then evaluate BLoB's generalization and uncertainty estimation abilities in both in-distribution (Sec. 4.2) and out-of-distribution scenarios (Sec. 4.3).

### 4.1 Settings

**Fine-tuning and Evaluation.** We implement BLoB in the PEFT library [63] and fine-tune the LlaMA2-7B [92] model on common-sense reasoning tasks. Following Laplace-LoRA [116], we apply LoRA to the output layer as well as the queries and values of all the attention layers. For hyperparameters, we strictly adhere to the default settings in the PEFT library and the original LoRA paper [63, 41] to ensure maximal reproducibility. This includes the number of training steps, learning rate, and LoRA rank $r$ (see Appendix B.1 for details). For common-sense reasoning tasks, we select

the next token logits corresponding to possible answers from each dataset and fine-tune the LLM to maximize the likelihood of the correct token. For evaluation, in addition to Accuracy (**ACC**), we use Expected Calibration Error (**ECE** [31]) and Negative Log-Likelihood (**NLL**) to assess the models' uncertainty estimation ability (see Appendix B.2 for details).

**Baselines and Implementation Details.** We compare **BLoB** with state-of-the-art uncertainty estimation methods applied to the LoRA adapters of LLMs, including Monte-Carlo Dropout (**MCD**) [29], Bayes By Backprop (**BBB**) [11], Deep Ensemble (**ENS**) [51, 8, 103], and the latest Laplace-LoRA (**LAP**) [116]. We also report the performance of two standard PEFT baseline methods for reference: Maximum Likelihood Estimation (**MLE**) [41] and Maximum A Posteriori (**MAP**).

For MLE, we use the LoRA implementation. For MAP, we use a weight decay rate of $1e-5$. For MCD, we use an ensemble of 10 LoRAs with a dropout rate of $p = 0.1$. For ENS, we independently fine-tune 3 LoRAs and average their logits during evaluation. For BBB, we adopt the default settings from the Bayesian-Torch library [48] and only Bayesianize the $A$ matrix, similar to BLoB. We sample $N = 10$ times for BBB during test. We re-implement LAP and apply it to the MAP checkpoints. We keep all BLoB-specific hyperparameters consistent across all datasets. Typically, we set the number of samples $K = 1$ during training for all our BLoB experiments, which highlights BLoB's sampling efficiency. As shown in Table 1, we also report BLoB's performance with different numbers of samples during Bayesian inference, where $N = 0$ indicates directly using the mean of the weight distribution for prediction.

## 4.2 Results on In-distribution Datasets

We fine-tune Llama2-7B on six common-sense reasoning tasks: Winogrande-small (WG-S), Winogrande-medium (WG-M) [82], ARC-Challenge (ARC-C) [18], ARC-Easy (ARC-E) [18], Open-BookQA (OBQA) [65], and BoolQ [17]. For all baseline methods, using the same pre-trained LLM backbone, we maintain consistent hyperparameters across all datasets and do not use additional validation sets to achieve higher performance (See Appendix B.3 for detailed settings).

Table 1 shows the performance of BLoB compared to the baselines, including ACC, ECE, and NLL, on the in-distribution test set with the pre-trained Llama2-7B model. The high ECE and NLL for MLE indicate overconfidence in LLMs during conventional fine-tuning, except for BoolQ due to its large dataset size. Simple but popular baselines like MAP, MCD, and ENS show mixed results in terms of NLL and/or ECE, highlighting the challenge of uncertainty estimation during LLM fine-tuning. LAP, the most competitive post-training baseline for uncertainty estimation, significantly reduces NLL and ECE on some datasets but lacks consistent performance, as indicated by its failures on ARC-C and ARC-E. BBB mitigates the overconfidence issue in LLMs across almost all datasets, showcasing the advantage of jointly optimizing the mean and covariance of the variational weight distributions during fine-tuning. However, there remains considerable room for improvement.

BLoB consistently achieves better or comparable performance across all datasets. With the number of samples during inference set to $N = 10$, the same as MCD, BLoB provides the best uncertainty estimation performance, significantly reducing NLL and ECE, and greatly mitigating overconfidence while maintaining comparable or better ACC than MLE. Even with half the number of samples, $N = 5$, BLoB still delivers performance comparable to that of $N = 10$ and outperforms other baselines on most datasets. By abandoning the modeling of the posterior distribution, prediction using the mean of the weight distribution, i.e., BLoB (N=0) sacrifices some degree of calibration in exchange for improved accuracy. Appendix C.5 presents the trade-off between accuracy and calibration, which is controlled by the standard deviation of the prior Gaussian distribution).

Besides Llama2-7B, we also include additional results for RoBERTa-base [60] on text classification tasks in Appendix C.1. Our method consistently achieved either the best or runner-up performance across nearly all datasets, demonstrating its versatility across different architectures.

---

[1] LAP encounters training failures on the ARC-C dataset under a unified setting, where the same hyper-parameter configuration is applied across all datasets. To achieve competitive performance with LAP on the ARC-C dataset, we deviate from this unified setting, allowing dataset-specific hyperparameters for LAP. Note that this introduces an unfair advantage for LAP. Specifically, for the LAP on ARC-C dataset, we set the dropout rate to 0.1, the learning rate to 5e-5, and applied early stopping at the 5,000-th iteration (out of 10,000).

Table 2: **Performance on in-distribution and out-of-distribution datasets**. All the uncertainty estimation methods are applied to the LoRA adapter added upon the pre-trained Llama2-7B weights.

| Metric | Method | In-Dist. | Smaller Dist. Shift | | Larger Dist. Shift | |
|---|---|---|---|---|---|---|
| | | OBQA [65] | ARC-C [18] | ARC-E [18] | Chem [38, 37] | Phy [38, 37] |
| ACC (↑) | MLE | $81.52_{\pm0.25}$ | $66.20_{\pm0.87}$ | $75.12_{\pm0.85}$ | $40.62_{\pm2.25}$ | $28.82_{\pm1.30}$ |
| | MAP | $81.38_{\pm0.91}$ | $\underline{69.59}_{\pm0.33}$ | $75.47_{\pm0.73}$ | $\mathbf{44.79}_{\pm0.00}$ | $28.47_{\pm1.20}$ |
| | MCD [29] | $81.72_{\pm0.10}$ | $69.03_{\pm0.70}$ | $76.00_{\pm1.58}$ | $42.71_{\pm0.01}$ | $29.17_{\pm4.54}$ |
| | ENS [51, 8, 103] | $81.38_{\pm0.65}$ | $67.34_{\pm0.70}$ | $75.18_{\pm2.03}$ | $43.75_{\pm1.04}$ | $30.56_{\pm2.62}$ |
| | BBB [11] | $82.06_{\pm0.59}$ | $67.25_{\pm1.18}$ | $75.83_{\pm0.75}$ | $42.36_{\pm0.49}$ | $30.21_{\pm2.25}$ |
| | LAP [116] | $\underline{82.12}_{\pm0.67}$ | $69.14_{\pm1.15}$ | $74.94_{\pm0.96}$ | $\underline{44.10}_{\pm1.30}$ | $31.60_{\pm0.49}$ |
| | BLoB (N=0) | $\mathbf{82.73}_{\pm0.41}$ | $\mathbf{69.93}_{\pm1.20}$ | $\mathbf{76.88}_{\pm0.41}$ | $41.67_{\pm2.25}$ | $31.94_{\pm1.77}$ |
| | BLoB (N=5) | $81.79_{\pm0.94}$ | $68.36_{\pm1.39}$ | $75.82_{\pm1.15}$ | $40.62_{\pm3.07}$ | $\mathbf{32.64}_{\pm0.98}$ |
| | BLoB (N=10) | $81.52_{\pm0.74}$ | $67.71_{\pm1.13}$ | $\underline{76.37}_{\pm0.80}$ | $\mathbf{44.79}_{\pm1.47}$ | $31.60_{\pm2.73}$ |
| ECE (↓) | MLE | $12.55_{\pm0.46}$ | $22.20_{\pm0.39}$ | $16.47_{\pm0.86}$ | $21.72_{\pm0.30}$ | $29.60_{\pm1.29}$ |
| | MAP | $15.34_{\pm0.27}$ | $19.31_{\pm1.46}$ | $15.68_{\pm0.51}$ | $17.55_{\pm1.95}$ | $30.25_{\pm2.18}$ |
| | MCD [29] | $14.45_{\pm0.84}$ | $19.54_{\pm0.33}$ | $15.32_{\pm1.16}$ | $17.9_{\pm0.63}$ | $29.53_{\pm4.20}$ |
| | ENS [51, 8, 103] | $13.26_{\pm0.82}$ | $\underline{7.59}_{\pm1.43}$ | $6.44_{\pm0.83}$ | $12.04_{\pm4.57}$ | $\underline{17.52}_{\pm1.28}$ |
| | BBB [11] | $11.38_{\pm1.07}$ | $19.90_{\pm0.66}$ | $13.41_{\pm0.85}$ | $15.67_{\pm1.23}$ | $26.10_{\pm4.76}$ |
| | LAP [116] | $8.70_{\pm1.77}$ | $\mathbf{5.84}_{\pm0.64}$ | $8.51_{\pm1.06}$ | $\underline{10.76}_{\pm3.41}$ | $\mathbf{13.91}_{\pm0.90}$ |
| | BLoB (N=0) | $8.36_{\pm0.38}$ | $14.00_{\pm1.02}$ | $10.70_{\pm0.39}$ | $15.05_{\pm0.77}$ | $22.90_{\pm2.27}$ |
| | BLoB (N=5) | $\mathbf{3.40}_{\pm0.57}$ | $9.76_{\pm0.71}$ | $\underline{5.96}_{\pm0.93}$ | $14.33_{\pm1.55}$ | $18.15_{\pm1.96}$ |
| | BLoB (N=10) | $\underline{3.77}_{\pm1.47}$ | $9.55_{\pm0.40}$ | $\mathbf{5.48}_{\pm1.27}$ | $\mathbf{9.77}_{\pm1.35}$ | $18.29_{\pm1.35}$ |
| NLL (↓) | MLE | $0.73_{\pm0.03}$ | $1.16_{\pm0.00}$ | $0.92_{\pm0.03}$ | $1.56_{\pm0.06}$ | $1.66_{\pm0.05}$ |
| | MAP | $1.06_{\pm0.04}$ | $1.10_{\pm0.07}$ | $0.93_{\pm0.04}$ | $1.55_{\pm0.06}$ | $1.65_{\pm0.03}$ |
| | MCD [29] | $1.06_{\pm0.08}$ | $1.08_{\pm0.01}$ | $0.88_{\pm0.03}$ | $1.59_{\pm0.07}$ | $1.67_{\pm0.05}$ |
| | ENS [51, 8, 103] | $0.75_{\pm0.01}$ | $0.86_{\pm0.01}$ | $0.69_{\pm0.03}$ | $\mathbf{1.28}_{\pm0.00}$ | $\underline{1.39}_{\pm0.03}$ |
| | BBB [11] | $0.66_{\pm0.05}$ | $1.06_{\pm0.01}$ | $0.79_{\pm0.02}$ | $1.49_{\pm0.05}$ | $1.62_{\pm0.06}$ |
| | LAP [116] | $\underline{0.52}_{\pm0.01}$ | $\mathbf{0.81}_{\pm0.00}$ | $0.70_{\pm0.02}$ | $\underline{1.35}_{\pm0.03}$ | $\mathbf{1.36}_{\pm0.01}$ |
| | BLoB (N=0) | $0.56_{\pm0.02}$ | $0.89_{\pm0.02}$ | $0.67_{\pm0.02}$ | $1.44_{\pm0.00}$ | $1.53_{\pm0.02}$ |
| | BLoB (N=5) | $0.53_{\pm0.01}$ | $0.85_{\pm0.00}$ | $\underline{0.64}_{\pm0.01}$ | $1.39_{\pm0.02}$ | $1.48_{\pm0.01}$ |
| | BLoB (N=10) | $\mathbf{0.50}_{\pm0.01}$ | $\underline{0.83}_{\pm0.01}$ | $\mathbf{0.60}_{\pm0.01}$ | $1.38_{\pm0.01}$ | $1.46_{\pm0.02}$ |

## 4.3 Results on Out-of-Distribution Datasets

We use models fine-tuned on OBQA [65] to evaluate the generalization ability of different methods under distributional shifts. OBQA consists of multiple-choice elementary-level science questions. We categorize the distributional shifts into two types: *smaller* and *larger* shifts. The ARC [18] dataset, which also consists of multiple-choice science questions, represents a smaller distributional shift. The college-level chemistry and physics subsets of MMLU [38] represent larger distributional shifts.

The results in Table 2 highlight BLoB's superior OOD generalization ability compared to other methods on both smaller and larger distribution shifts. BLoB achieves the highest accuracy when solely utilizing the mean of the weight distribution in smaller distribution shifts. For larger distribution shifts, incorporating uncertainty through sampling improves model accuracy. Regarding uncertainty estimation, BLoB demonstrates the best or second-best performance in smaller distribution shifts. Although there is a slight performance drop with larger distribution shifts, BLoB remains comparable to baselines such as ENS and LAP.

## 5 Related Work

**Parameter-Efficient Fine-Tuning (PEFT) for LLMs.** Due to the prohibitively large size of LLMs, parameter-efficient fine-tuning has become a trending topic. Computational paradigms in this area include adapter-based fine-tuning [40, 36, 80, 75, 62], prompt-based fine-tuning [34, 54, 59, 57, 94, 6], and partial fine-tuning [119, 124, 5, 112, 32]. Among these, LoRA [41] has gained significant attention due to its simplicity and effectiveness. Building on LoRA, numerous studies have aimed to further optimize parameter efficiency when fine-tuning large models [26, 35, 22, 21]. For instance, KronA models weight updates as the Kronecker product of two smaller matrices without decreasing the update rank [26], and SVDiff performs Singular Value Decomposition (SVD) on the original weight matrices, fine-tuning only the singular values [35]. However, in this paper, we focus solely on

Bayesianizing LoRA due to its widespread application in existing works. We also note that BLoB can be naturally adapted to handle different LoRA variants.

**Uncertainty Estimation in Large Language Models.** Large-scale pre-trained models are well-calibrated during pre-training [44], but fail to accurately express predictive uncertainty during inference [3, 107, 44, 43, 87], especially after fine-tuning [8, 103, 116, 69]. This indicates that measures effective during pre-training [93, 114, 16, 122, 14] may lose their power of uncertainty estimation after fine-tuning for domain-specific knowledge. To address this issue, [27, 123] define priors and approximate posteriors on the full attention weights during fine-tuning, achieving better uncertainty estimation but at a significant cost in time and space. Consequently, recent work integrates Bayesian methods and PEFT for efficient uncertainty estimation. For instance, [8, 103] train and store multiple copies of different LoRAs, ensembling their outputs during inference to achieve somewhat better results. [116] applies Kronecker factorized Laplace approximation on fine-tuned LoRA. However, such post-training procedures bifurcate posterior approximation into two stages, leading to suboptimal estimation. In contrast, our BLoB enables simultaneous estimation of both the mean and covariance of LLM parameters in a single fine-tuning stage, substantially improving performance.

# 6 Conclusion

In this work, we propose a principled Bayesianization framework for parameter-efficiently fine-tuning LLMs. Our theoretical analysis shows that a full-weight variational distribution can be efficiently optimized by approximately using a low-rank space of the weight update matrices. Our empirical evaluations corroborate this theoretical insight, demonstrating superior generalization and uncertainty estimation capabilities across diverse scenarios compared to various baseline methods. Building on LoRA, our approach seamlessly integrates with existing LLM architectures while imposing minimal additional memory overhead and training time. Our method highlights that jointly learning the mean and covariance of the variational distribution during fine-tuning can mutually improve both, underscoring the powerful potential of Bayesian methods in enhancing the reliability and generalization of LLMs.

# 7 Limitations

The main limitations of our proposed BLoB method are: (i) BLoB is confined to fine-tuning scenarios and is not applicable to training-free tasks, such as direct uncertainty estimation during inference [64]. (ii) As a typical mean-field variational inference method, BLoB requires multiple sampling iterations during inference, which challenges stable and efficient deployment. (iii) While BLoB's effectiveness has been empirically demonstrated for downstream classification tasks, its application to generation tasks requires further investigation.

## Acknowledgement

The authors thank the reviewers/AC for the constructive comments to improve the paper. We thank Sanket Jantre for identifying improvements to our proof and for other valuable discussions. HS and HW are partially supported by Microsoft Research AI & Society Fellowship, NSF Grant IIS-2127918, NSF CAREER Award IIS-2340125, NIH Grant 1R01CA297832, and the Amazon Faculty Research Award. This research is also supported by NSF National Artificial Intelligence Research Resource (NAIRR) Pilot. The views and conclusions contained herein are those of the authors and should not be interpreted as necessarily representing the official policies, either expressed or implied, of the sponsors.

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

# Appendix

In Appendix A, we present the proofs for the theorems in the main body of our paper. In Appendix B, we introduce the experimental settings, including evaluation metrics and training schemes. Finally, in Appendix C, we present supplementary empirical results including the experiments on another language model and analysis of the space and time cost of our algorithm.

## A  Proof of Theorems and Claims

In this section, we first present the proof of the two main theorems (Theorem 3.1 and Theorem 3.2) in Appendix A.1. Next, we show how a analysis on our design of parameterization in Appendix A.2. Finally, we provide a detailed derivation of the LoRA Flipout in Appendix A.3.

### A.1  Proof of Main Theorems

**Theorem 3.1** (**Variational Distribution of the Full-Weight Matrix in BLoB**). *With the pre-trained weight matrix $\boldsymbol{W}_0 \in \mathbb{R}^{m \times n}$ and the low-rank weight update matrix $\boldsymbol{B} \in \mathbb{R}^{m \times r}$, suppose that the variational distribution of the other low-rank update matrix $\boldsymbol{A} \in \mathbb{R}^{r \times n}$ is Gaussian with $q(\boldsymbol{A}|\boldsymbol{\theta} = \{\boldsymbol{M}, \boldsymbol{\Omega}\}) = \prod_{ij} \mathcal{N}(A_{ij}|M_{ij}, \Omega_{ij}^2)$, where $\boldsymbol{M} = [M_{ij}] \in \mathbb{R}^{r \times n}$ and $\boldsymbol{\Omega} = [\Omega_{ij}] \in \mathbb{R}^{r \times n}$ are its mean and standard deviation, respectively. The equivalent variational distribution defined on the full weight matrix $\boldsymbol{W}$ as in Eqn. 3 is given by*

$$q(\text{vec}(\boldsymbol{W})|\boldsymbol{B}, \boldsymbol{\theta}) = \mathcal{N}(\text{vec}(\boldsymbol{W})|\boldsymbol{\mu}_q, \boldsymbol{\Sigma}_q),$$
$$\text{where} \quad \boldsymbol{\mu}_q = \text{vec}(\boldsymbol{W}_0 + \boldsymbol{B}\boldsymbol{M}),$$
$$\boldsymbol{\Sigma}_q = [\boldsymbol{I}_n \otimes \boldsymbol{B}] \cdot [\text{diag}(\text{vec}(\boldsymbol{\Omega})^2)] \cdot [\boldsymbol{I}_n \otimes \boldsymbol{B}^\top].$$

*Proof.* We begin by calculating the mean value of $q$,

$$\boldsymbol{\mu}_q = \text{vec}(\mathbb{E}[\boldsymbol{W}_0 + \boldsymbol{B}\boldsymbol{A}]) \tag{15}$$
$$= \text{vec}(\boldsymbol{W}_0 + \boldsymbol{B}\mathbb{E}[\boldsymbol{A}]) \tag{16}$$
$$= \text{vec}(\boldsymbol{W}_0 + \boldsymbol{B}\boldsymbol{M}). \tag{17}$$

Suppose the deterministic matrix $\boldsymbol{B} = [\boldsymbol{b}_1, \boldsymbol{b}_2, \cdots, \boldsymbol{b}_r] \in \mathbb{R}^{m \times r}$, random matrix $\boldsymbol{A} = [\boldsymbol{a}_1, \boldsymbol{a}_2, \cdot, \boldsymbol{a}_r]^\top \in \mathbb{R}^{r \times n}$, with its underlying parameters of mean and standard deviation defined likewise $\boldsymbol{M} \in \mathbb{R}^{r \times n}$ and $\boldsymbol{\Omega} \in \mathbb{R}^{r \times n}$. We have $\boldsymbol{W} = \boldsymbol{B}\boldsymbol{A} = \sum_{i=1}^r \boldsymbol{b}_i \cdot \boldsymbol{a}_i^\top$. We then rewrite $\text{vec}(\boldsymbol{W})$ in the form of Kronecker product $\otimes$:

$$\text{vec}(\boldsymbol{W}) = \text{vec}(\sum_{i=1}^r \boldsymbol{b}_i \cdot \boldsymbol{a}_i^\top) = \sum_{i=1}^r (\boldsymbol{a}_i \otimes \boldsymbol{b}_i) \tag{18}$$

We then calculate the covariance matrix $\boldsymbol{\Sigma}_q$ as

$$\boldsymbol{\Sigma}_q = \text{cov}[\text{vec}(\boldsymbol{W}), \text{vec}(\boldsymbol{W})] = \text{cov}[\sum_{i=1}^r (\boldsymbol{a}_i \otimes \boldsymbol{b}_i), \sum_{i=1}^r (\boldsymbol{a}_i \otimes \boldsymbol{b}_i)] \tag{19}$$

$$= \sum_{i=1}^r \text{cov}[\boldsymbol{a}_i \otimes \boldsymbol{b}_i, \boldsymbol{a}_i \otimes \boldsymbol{b}_i] + \sum_{i \neq j} \text{cov}[\boldsymbol{a}_i \otimes \boldsymbol{b}_i, \boldsymbol{a}_j \otimes \boldsymbol{b}_j] \tag{20}$$

$$= \sum_{i=1}^r \left\{ \mathbb{E}_{\boldsymbol{a}_i}[(\boldsymbol{a}_i \otimes \boldsymbol{b}_i)(\boldsymbol{a}_i \otimes \boldsymbol{b}_i)^\top] - \mathbb{E}_{\boldsymbol{a}_i}[(\boldsymbol{a}_i \otimes \boldsymbol{b}_i)]\mathbb{E}_{\boldsymbol{a}_i}[(\boldsymbol{a}_i \otimes \boldsymbol{b}_i)^\top] \right\} \tag{21}$$

$$= \sum_{i=1}^r \left\{ \mathbb{E}_{\boldsymbol{a}_i}[(\boldsymbol{a}_i \boldsymbol{a}_i^\top)] \otimes (\boldsymbol{b}_i \boldsymbol{b}_i^\top) - (\mathbb{E}_{\boldsymbol{a}_i}[\boldsymbol{a}_i]\mathbb{E}_{\boldsymbol{a}_i}[\boldsymbol{a}_i]^\top) \otimes (\boldsymbol{b}_i \boldsymbol{b}_i^\top) \right\} \tag{22}$$

$$= \sum_{i=1}^r \text{diag}(\boldsymbol{\sigma}_i^2) \otimes (\boldsymbol{b}_i \boldsymbol{b}_i^\top) \tag{23}$$

$$= [\boldsymbol{I}_n \otimes \boldsymbol{B}] \cdot [\text{diag}(\text{vec}(\boldsymbol{\Omega}^2))] \cdot [\boldsymbol{I}_n \otimes \boldsymbol{B}^\top], \tag{24}$$

completing the proof. $\qquad\square$

It is crucial to note here, the final covariance matrix of $q(\text{vec}(\boldsymbol{W}))$ follows a block-diagonal structure, which will be further utilized for the proof of Theorem 3.2. Defining $\boldsymbol{\Sigma}_i = \text{diag}(\boldsymbol{\Omega}_{i:}^2)$, we have:

$$\boldsymbol{\Sigma}_q = \begin{bmatrix} \boldsymbol{B}\boldsymbol{\Sigma}_1\boldsymbol{B}^\top & & \\ & \ddots & \\ & & \boldsymbol{B}\boldsymbol{\Sigma}_n\boldsymbol{B}^\top \end{bmatrix}. \tag{25}$$

Another important fact about $\boldsymbol{\Sigma}_q$ is its singularity. It can be seen directly as we consider the rank of any one of the block matrix $\boldsymbol{B}\boldsymbol{\Sigma}_i\boldsymbol{B}^\top \in \mathbb{R}^{m \times m}, \forall i \in [n]$:

$$\text{r}(\boldsymbol{B}\boldsymbol{\Sigma}\boldsymbol{B}^\top) \leq \min\{\text{r}(\boldsymbol{B}), \text{r}(\boldsymbol{\Sigma}_i), \text{r}(\boldsymbol{B}^\top)\} \leq r < m, \tag{26}$$

where $r$ is the rank of LoRA, strictly smaller than the output dimension of $m$.

**Theorem 3.2** (**Efficient Computation of Full-Weight KL Divergence**). *Suppose the pre-trained weights $\boldsymbol{W}_0$, update matrix $\boldsymbol{B}$, and the variational distribution $q(\boldsymbol{A}|\boldsymbol{\theta})$ are defined as in Theorem 3.1, and the prior distribution of the full-weight matrix $P(\text{vec}(\boldsymbol{W}))$ is defined as Eqn. 9. Consider the Gaussian prior distribution $P(\boldsymbol{A}) = \prod_{ij} \mathcal{N}(A_{ij}|0, \sigma_p^2)$; we then have:*

$$\text{KL}[q(\text{vec}(\boldsymbol{W})|\boldsymbol{B}, \boldsymbol{\theta})\|P(\text{vec}(\boldsymbol{W}))] = \text{KL}[q(\boldsymbol{A}|\boldsymbol{\theta})\|P(\boldsymbol{A})],$$

*if $\widetilde{\boldsymbol{R}} = [\sigma_p\boldsymbol{I}_n \otimes \boldsymbol{R}]$, where $\boldsymbol{R}$ satisfies $\boldsymbol{R}\boldsymbol{R}^\top = \boldsymbol{B}\boldsymbol{B}^\top$.*

*Proof.* We start by assuming the low-rank structure of the prior $P(\text{vec}(\boldsymbol{W}))$, and then reveal the conditions reaching to our final conclusion step by step.

Typically, for two Gaussian distributions $q$ and $p$ whose covariance matrices $\boldsymbol{\Sigma}_q \in \mathbb{R}^{d \times d}$ and $\boldsymbol{\Sigma}_p \in \mathbb{R}^{d \times d}$ are both full-rank, and their means as $\boldsymbol{\mu}_q \in \mathbb{R}^d$ and $\boldsymbol{\mu}_p \in \mathbb{R}^d$, we have their KL-divergence as

$$\text{KL}[q\|p] = \frac{1}{2}\left[\log\frac{|\boldsymbol{\Sigma}_p|}{|\boldsymbol{\Sigma}_q|} - d + \text{tr}(\boldsymbol{\Sigma}_p^{-1}\boldsymbol{\Sigma}_q) + (\boldsymbol{\mu}_q - \boldsymbol{\mu}_p)^\top\boldsymbol{\Sigma}_p^{-1}(\boldsymbol{\mu}_q - \boldsymbol{\mu}_p)\right]. \tag{27}$$

The singularity of the covariance matrices of $P(\text{vec}(\boldsymbol{W}))$ and $q(\text{vec}(\boldsymbol{W}))$, i.e., $|\boldsymbol{\Sigma}_q| = |\boldsymbol{\Sigma}_p| = 0$, can cause issues when computing the KL-divergence as it includes the log-determinant term. Therefore in this proof, we consider the alternative of the covariance matrices, where an extremely small diagonal elements are added.

**For the prior distribution**, following the alternative form of a degenerate Gaussian [83], as suggested in Eqn. 9, we assume

$$P(\text{vec}(\boldsymbol{W})) = \mathcal{N}(\boldsymbol{W}_0, \boldsymbol{\Sigma}_p), \tag{28}$$
$$\text{where} \qquad \boldsymbol{\Sigma}_p = \lambda\boldsymbol{I} + \widetilde{\boldsymbol{R}}\widetilde{\boldsymbol{R}}^\top, \quad (\lambda \to 0^+).$$

By default, we assume that the low-rank tall matrix $\widetilde{\boldsymbol{R}} \in \mathbb{R}^{(mn) \times r'}$ has the full column rank $r'$. Otherwise if $\text{r}(\widetilde{\boldsymbol{R}}) = r'' < r'$, then we can in effect consider a new matrix component $\widetilde{\boldsymbol{R}}' \in \mathbb{R}^{(mn) \times r''}$ that has the same rank as $r''$, which satisfies our assumption of full column rank. Therefore, we have the SVD decomposition of $\widetilde{\boldsymbol{R}}$ is given by

$$\widetilde{\boldsymbol{R}} = \boldsymbol{U}_R\boldsymbol{D}_R\boldsymbol{V}_R^\top, \tag{29}$$

where $\boldsymbol{U}_R \in \mathbb{R}^{(mn) \times (mn)}$ and $\boldsymbol{V}_R \in \mathbb{R}^{r' \times r'}$ are orthonormal, i.e., $\boldsymbol{U}_R\boldsymbol{U}_R^\top = \boldsymbol{U}_R^\top\boldsymbol{U}_R = \boldsymbol{I}_{(mn)}$ and $\boldsymbol{V}_R\boldsymbol{V}_R^\top = \boldsymbol{V}_R^\top\boldsymbol{V}_R = \boldsymbol{I}_{r'}$. $\boldsymbol{D}_R$ is a tall matrix where its upper part is diagonal and the lower part is a zero matrix, denoted as $\boldsymbol{D}_R = [\boldsymbol{D}_R^*, \boldsymbol{O}]^\top = [\text{diag}([d_{R_1} > 0, d_{R_2} > 0, \cdots, d_{R_{r'}} > 0]), \boldsymbol{O}]^\top$.

**For the approximate posterior** $q(\text{vec}(\boldsymbol{W})|\boldsymbol{B}, \boldsymbol{\theta})$, we consider

$$q(\text{vec}(\boldsymbol{W})|\boldsymbol{B}, \boldsymbol{\theta}) = \mathcal{N}(\boldsymbol{W}_0 + \boldsymbol{B}\boldsymbol{M}, \boldsymbol{\Sigma}_q), \tag{30}$$
$$\text{where} \qquad \boldsymbol{\Sigma}_q = \lambda\boldsymbol{I} + \widetilde{\boldsymbol{B}}\boldsymbol{\Sigma}\widetilde{\boldsymbol{B}}^\top, \quad (\lambda \to 0^+),$$

where we simplify the notation for the covariance matrix $\boldsymbol{\Sigma}_q$ by defining $\widetilde{\boldsymbol{B}} = [\boldsymbol{I}_n \otimes \boldsymbol{B}] \in \mathbb{R}^{(mn) \times (mr)}$ and $\boldsymbol{\Sigma} = \text{diag}(\text{vec}(\boldsymbol{\Omega})^2)$. Likewise, we have the SVD-decomposed matrices for $\widetilde{\boldsymbol{B}}$ where they are defined in the same way as Eqn. 29:

$$\widetilde{\boldsymbol{B}} = \boldsymbol{U}_B\boldsymbol{D}_B\boldsymbol{V}_B^\top, \tag{31}$$

where $\boldsymbol{U}_B$ and $\boldsymbol{V}_B$ are orthogonal matrices, and $\boldsymbol{D}_B = [\boldsymbol{D}_B^*, \boldsymbol{O}]^\top = [\mathrm{diag}([d_{B_1} > 0, d_{B_2} > 0, \cdots, d_{B_{mr}} > 0]), \boldsymbol{O}]^\top$.

First, we calculate the log-determinant part of the KL-divergence. For the log-determinant of the covariance matrix of the prior distribution $\boldsymbol{\Sigma}_p$, by applying SVD decomposition in Eqn. 29, we have

$$\log|\boldsymbol{\Sigma}_p| = \log|\lambda\boldsymbol{I} + \widetilde{\boldsymbol{R}}\widetilde{\boldsymbol{R}}^\top| = \log|\lambda\boldsymbol{I} + \boldsymbol{U}_R\boldsymbol{D}_R\boldsymbol{V}_R^\top\boldsymbol{V}_R\boldsymbol{D}_R^\top\boldsymbol{U}_R^\top| \tag{32}$$

$$= \log|\boldsymbol{U}_R(\lambda\boldsymbol{I} + \boldsymbol{D}_R\boldsymbol{D}_R^\top)\boldsymbol{U}_R^\top| \tag{33}$$

$$= \log\left|\boldsymbol{U}_R \begin{bmatrix} (\boldsymbol{D}_R^*)^2 + \lambda\boldsymbol{I}_{r'} & \boldsymbol{O} \\ \boldsymbol{O} & \lambda\boldsymbol{I}_{mn-r'} \end{bmatrix} \boldsymbol{U}_R^\top\right| \tag{34}$$

$$= \log|(\boldsymbol{D}_R^*)^2 + \lambda\boldsymbol{I}_{r'}| + \log|\lambda\boldsymbol{I}_{mn-r'}| \tag{35}$$

$$= (mn - r')\log\lambda + \sum_{i=1}^{r'}\log(d_{R_i}^2 + \lambda). \tag{36}$$

Following (almost) the same idea, we now have the log-determinant of the variational distribution's covariance $\boldsymbol{\Sigma}_q$ as

$$\log|\boldsymbol{\Sigma}_q| = \log|\lambda\boldsymbol{I} + \widetilde{\boldsymbol{B}}\widetilde{\boldsymbol{B}}^\top| = \log\left|\begin{matrix} \boldsymbol{D}_B^*\boldsymbol{V}_B^\top\boldsymbol{\Sigma}\boldsymbol{V}_B\boldsymbol{D}_B^* + \lambda\boldsymbol{I}_{mr} & \boldsymbol{O} \\ \boldsymbol{O} & \lambda\boldsymbol{I}_{mn-mr} \end{matrix}\right| \tag{37}$$

$$= (mn - mr)\log\lambda + 2\log|\boldsymbol{D}_B^*| + \log|\boldsymbol{V}_B^\top\boldsymbol{\Sigma}\boldsymbol{V}_B + \lambda(\boldsymbol{D}_B^*)^{-2}| \tag{38}$$

$$= (mn - mr)\log\lambda + 2\sum_{i=1}^{mr}\log d_{B_i} + \log|\boldsymbol{\Sigma}| + \log|\boldsymbol{I} + \lambda\boldsymbol{V}_B^\top\boldsymbol{\Sigma}^{-1}\boldsymbol{V}_B(\boldsymbol{D}_B^*)^{-2}|. \tag{39}$$

We make two observations when $\lambda \to 0^+$: (i) compare the terms that contain $\log\lambda$ on both sides, to make sure the log-determinant in the divergence term *bounded*, we have to set $r' = mr$; (ii) the last term in Eqn. 39, $\log|\boldsymbol{I} + \lambda\boldsymbol{V}_B^\top\boldsymbol{\Sigma}^{-1}\boldsymbol{V}_B(\boldsymbol{D}_B^*)^{-2}| = \log|\boldsymbol{I}| = 0$. Therefore, we have

$$\log\frac{|\boldsymbol{\Sigma}_p|}{|\boldsymbol{\Sigma}_q|} = \sum_{i=1}^{mr}\log\frac{d_{R_i}^2 + \lambda}{d_{B_i}^2} - \log|\boldsymbol{\Sigma}|. \tag{40}$$

Next we calculate $\mathrm{tr}(\boldsymbol{\Sigma}_p^{-1}\boldsymbol{\Sigma}_q)$ in Eqn. 27. Following the same assumptions and notations above, we have the inverse of the covariance of the prior distribution as

$$\boldsymbol{\Sigma}_p^{-1} = \boldsymbol{U}_R \begin{bmatrix} [(\boldsymbol{D}_R^*)^2 + \lambda\boldsymbol{I}]^{-1} & \boldsymbol{O} \\ \boldsymbol{O} & \lambda^{-1}\boldsymbol{I} \end{bmatrix} \boldsymbol{U}_R^\top. \tag{41}$$

Hence we have

$$\mathrm{tr}(\boldsymbol{\Sigma}_p^{-1}\boldsymbol{\Sigma}_q) = \mathrm{tr}(\boldsymbol{U}_R \begin{bmatrix} [(\boldsymbol{D}_R^*)^2 + \lambda\boldsymbol{I}]^{-1} & \boldsymbol{O} \\ \boldsymbol{O} & \lambda^{-1}\boldsymbol{I} \end{bmatrix} \boldsymbol{U}_R^\top\boldsymbol{U}_B \begin{bmatrix} \boldsymbol{D}_B^*\boldsymbol{V}_B^\top\boldsymbol{\Sigma}\boldsymbol{V}_B\boldsymbol{D}_B^* + \lambda\boldsymbol{I} & \boldsymbol{O} \\ \boldsymbol{O} & \lambda\boldsymbol{I} \end{bmatrix} \boldsymbol{U}_B^\top). \tag{42}$$

By using the condition $\boldsymbol{R}\boldsymbol{R}^\top = \boldsymbol{B}\boldsymbol{B}^\top$ and $\widetilde{\boldsymbol{R}} = [\sigma_p\boldsymbol{I}_n \otimes \boldsymbol{R}]$ defined in Theorem 3.2, we have

$$\widetilde{\boldsymbol{R}}\widetilde{\boldsymbol{R}}^\top = \sigma_p^2\widetilde{\boldsymbol{B}}\widetilde{\boldsymbol{B}}^\top, \tag{43}$$

and there exists an orthogonal matrix $\boldsymbol{P} \in \mathbb{R}^{(mr)\times(mr)}$, such that

$$\widetilde{\boldsymbol{R}} = \sigma_p\widetilde{\boldsymbol{B}}\boldsymbol{P}. \tag{44}$$

The SVD decomposition on $\widetilde{\boldsymbol{R}}$ can then be formulated as:

$$\widetilde{\boldsymbol{R}} = \sigma_p\boldsymbol{U}_B\boldsymbol{D}_B\boldsymbol{V}_B^\top\boldsymbol{P} \tag{45}$$

$$= (\boldsymbol{U}_R = \boldsymbol{U}_B)(\boldsymbol{D}_R = \sigma_p\boldsymbol{D}_B)(\boldsymbol{V}_R = \boldsymbol{V}_B^\top\boldsymbol{P}). \tag{46}$$

Substituting $\boldsymbol{U}_R, \boldsymbol{D}_R, \boldsymbol{V}_R$ back to Eqn. 42 and applying $\lambda \to 0^+$, we have

$$\mathrm{tr}(\boldsymbol{\Sigma}_p^{-1}\boldsymbol{\Sigma}_q) = \mathrm{tr}(\boldsymbol{I}_{mn-nr}) + \mathrm{tr}([(\sigma_p\boldsymbol{D}^*)^2 + \lambda\boldsymbol{I}]^{-1}[\boldsymbol{D}_B^*\boldsymbol{V}_B^\top\boldsymbol{\Sigma}\boldsymbol{V}_B\boldsymbol{D}_B^*]) \tag{47}$$

$$= (mn - nr) + \sigma_p^{-2}\mathrm{tr}(\boldsymbol{V}_B^\top\boldsymbol{\Sigma}\boldsymbol{V}_B) \tag{48}$$

$$= (mn - nr) + \sigma_p^{-2}\mathrm{tr}(\boldsymbol{\Sigma}). \tag{49}$$

For the quadratic term in Eqn. 27, the pre-trained weights $\boldsymbol{W}_0$ cancel out, and we can calculate it as

$$\text{vec}(\boldsymbol{BM})^\top \boldsymbol{\Sigma}_p^{-1} \text{vec}(\boldsymbol{BM}) \tag{50}$$

$$=[\boldsymbol{M}_{:1}^\top \boldsymbol{B}^\top, \cdots, \boldsymbol{M}_{:1}^\top \boldsymbol{B}^\top] \begin{bmatrix} (\boldsymbol{B\Sigma}_1 \boldsymbol{B}^\top)^{-1} & & \\ & \ddots & \\ & & (\boldsymbol{B\Sigma}_n \boldsymbol{B}^\top)^{-1} \end{bmatrix} \begin{bmatrix} \boldsymbol{BM}_{:1} \\ \vdots \\ \boldsymbol{BM}_{:n} \end{bmatrix} \tag{51}$$

$$=\sum_{i=1}^{n} \boldsymbol{M}_{:i}^\top \boldsymbol{B}^\top (\boldsymbol{B\Sigma}_i \boldsymbol{B}^\top)^{-1} \boldsymbol{BM}_{:i} \tag{52}$$

$$=\sum_{i=1}^{n} \boldsymbol{M}_{:i}^\top (\boldsymbol{V}_B \begin{bmatrix} \frac{d_{B_1}^2}{\sigma_p^2(d_{B_1}^2+\lambda)} & & \\ & \ddots & \\ & & \frac{d_{B_{mr}}^2}{\sigma_p^2(d_{B_{mr}}^2+\lambda)} \end{bmatrix} \boldsymbol{V}_B^\top) \boldsymbol{M}_{:i} \tag{53}$$

$$=\frac{1}{\sigma_p^2} \sum_{i=1}^{n} \boldsymbol{M}_{:i}^\top \boldsymbol{M}_{:i} \tag{54}$$

$$=\frac{1}{\sigma_p^2} \|\boldsymbol{M}\|_2^2. \tag{55}$$

Finally, proof is completed by combining Eqn. 40, Eqn. 49, and Eqn. 55. □

## A.2 Analysis on BLoB Parameterization

**General Analysis on Parameterization.** Consider a path of parameterization for a single variable:

$$\rho \to \sigma = f(\rho) \to \mathcal{L} = l(\sigma), \tag{56}$$

where $\rho$ is the real parameter we perform update on, $f$ is our parameterization choice for the variable $\sigma$, and $l$ represents the loss function we aim to minimize. When comparing two different parameterization methods, we consider the same initial conditions of $\sigma = \sigma_0$, and we assume the same step size $\eta$ on the real parameter $\rho$. To show the influence of the choice of parameterization, we calculate the decrease of the loss value by performing one step of gradient descent. First, by the chain rule, the gradient w.r.t. $\rho_0$ is calculated as

$$\frac{d\mathcal{L}}{d\rho}\Big|_{\rho_0} = \frac{d\mathcal{L}}{d\sigma}\Big|_{\sigma_0} \cdot \frac{d\sigma}{d\rho}\Big|_{\rho_0} = l'(\sigma_0) \cdot f'(\rho_0). \tag{57}$$

After one step of the gradient descent, we have $\rho_1$ as

$$\rho_1 = \rho_0 - \eta \cdot l'(\sigma_0) \cdot f'(\rho_0), \tag{58}$$

and the loss value decreased at $\rho_1$ can be approximated by the first-order Taylor expansion,

$$\Delta\mathcal{L} = l(f(\rho_0)) - l(f(\rho_1)) \tag{59}$$

$$= l(f(\rho_0)) - [l(f(\rho_0 - \eta \cdot l'(\sigma_0) \cdot f'(\rho_0)))] \tag{60}$$

$$\approx l(f(\rho_0)) - [l(f(\rho_0) - \eta \cdot l'(\sigma_0) \cdot (f'(\rho_0))^2)] \tag{61}$$

$$\approx \eta \cdot (l'(\sigma_0))^2 \cdot (f'(\rho_0))^2. \tag{62}$$

Since the initialization of the different parameterization variable $\rho_0$ is set to ensure the same initial condition of $\sigma_0$ for different $f$s, we can see that the amount the loss decreases by after one step of update is proportional to the squared gradient $\Delta\mathcal{L} \propto (l'(\sigma_0))^2 \cdot (f'(\rho))^2 = (d\mathcal{L}/d\rho)^2$.

**Parameterization with** $\log(1 + \exp(\cdot))$ **or** $(\cdot)^2$**?** Previous VBNs [11, 49] typically use a softplus function $\sigma_q = \log(1 + \exp(\rho))$ to parameterize the standard deviation. For a single element $\rho$, the derivative of the closed-form solution of the KL divergence in Eqn. 11 is calculated as

$$\frac{d\,\text{KL}}{d\rho} = -\frac{e^\rho}{(1+e^\rho)\log(1+e^\rho)} + \frac{e^\rho \log(1+e^\rho)}{\sigma_p^2(1+e^\rho)}. \tag{63}$$

Due to the fact that $\sigma_q$ is typically initialized to a small value close to 0 to ensure stable optimization of the likelihood cost term (e.g., $1e-3$), and in order to ensure that the model obtains good uncertainty,

$\sigma_p$ is usually set to a larger value close to 1 (e.g., $1e-1$). In this case, the derivative of $\rho$ in Eqn. 63 is almost always a constant $-1$, which, based on our previous analysis, leads to slow convergence for large $\sigma_p$ values when $\sigma_q$ is small.

Therefore, we parameterize $\sigma_q$ with quadratic function: $\sigma_q = \rho^2$. In this case, the derivative of the KL divergence with respect to $\rho$ in Eqn. 11 becomes:

$$\frac{\mathrm{d}\,\mathrm{KL}}{\mathrm{d}\rho} = -\frac{2}{\rho} + \frac{2\rho^3}{\sigma_p^2}. \tag{64}$$

Under the same initialization conditions, the derivative in Eqn. 64 is approximately of the order of $\rho^{-1}$, leading to rapid convergence towards larger $\sigma_p$ values when $\sigma_q$ is small. Building on this, we use SGD without momentum to optimize the complexity loss term, thereby achieving the natural convergence of $\sigma_q$.

**Visualization.** To visually demonstrate the differences in the convergence of KL divergence during training with these two parameterizations, we set $\sigma_p = 1$ and employ gradient descent to optimize the KL divergence. As introduced in B.1, in practical mini-batch gradient descent, the KL divergence is weighted by $1/$#mini-batches. Therefore, assuming there are 100 mini-batches, the learning rate is set to 0.01, which translates to an actual learning rate of $1e-4$ for $\rho$. We initialize $\sigma_q = 0.01$ for both parameterizations. The growth of $\sigma_q$ during the KL training, for $\sigma_q = \log(1 + e^\rho)$ and $\sigma_q = \rho^2$ is shown in Fig. 2. In the same setting, the softplus parameterization takes nearly 100,000 gradient steps to converge, while the square parameterization takes only about 5,000 gradient steps. This modification makes it suitable for our fine-tuning setup, where the number of gradient steps is relatively small.

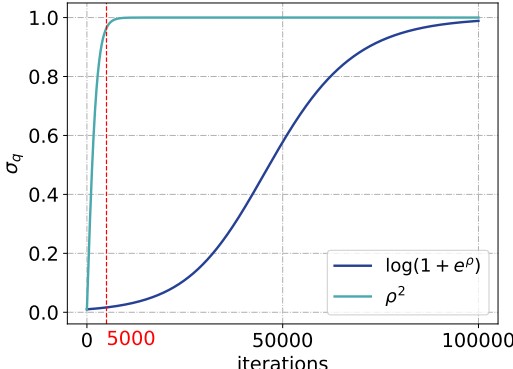

Figure 2: The growth curve of $\sigma_q = \log(1 + e^\rho)$ and $\sigma_q = \rho^2$ during the optimization of KL divergence (without data likelihood). The number of gradient steps (5000) is marked with the red line.

### A.3 Deriving Flipout for BLoB

For the $i$-th input vector $\boldsymbol{h}_i$ in a mini-batch, we randomly sample two flipping vector $\boldsymbol{s} \in \{-1, +1\}^n$ and $\boldsymbol{t} \in \{-1, +1\}^r$. Denoting $\boldsymbol{A}$ as the weight matrix sampled from posterior distribution, and $\Delta \boldsymbol{A}$ as the batched noise for sampling $\boldsymbol{A}$, the output vector $\boldsymbol{z}_i$ after applying flipout is:

$$\boldsymbol{z}_i = \boldsymbol{W}\boldsymbol{h}_i \tag{65}$$
$$= \boldsymbol{W}_0\boldsymbol{h}_i + \boldsymbol{B}\boldsymbol{A}\boldsymbol{h}_i \tag{66}$$
$$= \boldsymbol{W}_0\boldsymbol{h}_i + \boldsymbol{B}(\boldsymbol{M} + \Delta\boldsymbol{A})\boldsymbol{h}_i \tag{67}$$
$$= \boldsymbol{W}_0\boldsymbol{h}_i + \boldsymbol{B}\boldsymbol{M}\boldsymbol{h} + \boldsymbol{B}(\widehat{\boldsymbol{A}} \circ \boldsymbol{t}_i \boldsymbol{s}_i^\top)\boldsymbol{h}_i. \tag{68}$$

Similarly, for a mini-batch input matrix $\boldsymbol{H} \in \mathbb{R}^{n \times b}$ with batch size $b$, we randomly sample two low-rank flipping matrices $\boldsymbol{S} \in \{-1, +1\}^{n \times b}$ and $\boldsymbol{T} \in \{-1, +1\}^{b \times r}$. The batched output matrix $\boldsymbol{Z}$ after applying flipout is then:

$$\boldsymbol{Z} = \boldsymbol{W}_0\boldsymbol{H} + \boldsymbol{B}(\boldsymbol{M}\boldsymbol{H} + \left[\widehat{\boldsymbol{A}}(\boldsymbol{H} \circ \boldsymbol{S})\right] \circ \boldsymbol{T}) \tag{69}$$
$$= \boldsymbol{W}_0\boldsymbol{H} + \boldsymbol{B}(\boldsymbol{M}\boldsymbol{H} + [(\boldsymbol{E} \circ \boldsymbol{\Omega})(\boldsymbol{H} \circ \boldsymbol{S})] \circ \boldsymbol{T}). \tag{70}$$

| Hyperparameter | Model | |
| --- | --- | --- |
| | Roberta-base | Llama2-7B |
| Optimizer | AdamW | |
| LR Scheduler | Linear | |
| Warmup Ratio | 0.06 | |
| Learning Rate | $5e-4$ | $1e-4$ |
| Batch Size | 32 | 4 |
| Max Seq. Len. | 512 | 300 |
| LoRA $\alpha$ | 8 | 16 |
| LoRA $r$ | 8 | |

Table 3: Hyperparameters of LoRA

| Hyperparameter | Model | |
| --- | --- | --- |
| | Roberta-base | Llama2-7B |
| Optimizer of KL | SGD | |
| LR of KL | 0.002 | 0.01 |
| $\sigma_p$ | 0.2 | |
| $\epsilon$ | 0.05 | |
| $\gamma$ | 8 | |

Table 4: BLoB-Specific Hyperparameters

# B    Implementation Details

In this section, we first introduce the implementation details of BLoB in Appendix B.1, including the KL Re-weighting scheme, initialization of the parameters, and learning scheduling, etc. Next, we introduce the two evaluation metrics for uncertainty estimation in Appendix B.2. Finally, we present some statistics of the adopted datasets in Appendix B.3.

## B.1    Implementation of BLoB

**KL Re-weighting.** In mini-batch SGD, the training data $\mathcal{D}$ is randomly divided into $M$ equally sized subsets: $\mathcal{D}_1, \mathcal{D}_2, \ldots, \mathcal{D}_M$. For mini-batch $i = 1, 2, \ldots, M$, the cost function is:

$$\mathcal{F}(\mathcal{D}_i, \boldsymbol{\theta}) = -\mathbb{E}_{q(\boldsymbol{W}|\boldsymbol{\theta})}[\log P(\mathcal{D}_i|\boldsymbol{W})] + \lambda_i \, \mathrm{KL}[q(\boldsymbol{W}|\boldsymbol{\theta}) \parallel P(\boldsymbol{W})], \tag{71}$$

where $\lambda_i \in [0, 1]$ and $\sum_{i=1}^{M} \lambda_i = 1$. There are various approaches for controlling the weight of KL divergence. [30] utilizes $\lambda_i = 1/M$, while [11] adopts $\lambda_i = 2^{M-i}/2^M-1$. In fine-tuning tasks, we found that using a scheduler with $\lambda_i = 2^i/2^M-1$ performs well. This allows the model to find good fits to the data points within the early stages and then optimize the complexity cost in later stages.

In multiple epochs of mini-batch SGD, larger datasets require more iterations to complete one epoch, resulting in delayed convergence of the complexity cost. To enhance the stability of BLoB's performance across datasets with varying sizes, we pseudo-rescaled the size of the training dataset to make smaller datasets slightly larger and larger datasets slightly smaller. For the portions of the dataset that required expansion, we incorporated additional mini-batches from subsequent epochs. Conversely, for the datasets needing reduction, we deferred the excess mini-batches to subsequent epochs. We denote the size of original dataset as $L_0$, The rescaled dataset size is:

$$L^* = 100 \cdot L_0^{\frac{\pi}{\gamma}}, \tag{72}$$

where $\gamma$ is a coefficient used to control the scaling magnitude, and we set it to 8 in all experiments.

The pseudo-rescaling does not affect the likelihood cost in practical mini-batch gradient descent. In fact, it only changes the warm-up period in KL reweighting from $M$ to $L^*/\text{batch size}$, thereby facilitating more consistent optimization of the complexity cost across datasets of different sizes.

**Additional Details.** We initialize standard deviation parameterization matrix $\boldsymbol{G}$ by element-wise sampling from a uniform distribution with a range of $[\frac{\epsilon}{\sqrt{2}}, \epsilon]$, while keeping the remaining initialization settings consistent with LoRA. To maintain consistency, we use the same learning rate scheduler and warmup ratio for the optimizer of the KL term as we do for the likelihood term. We sample only once during the training process. During inference, we sample $N$ times, then take the average of the logits obtained after passing through the softmax function. Detailed hyperparameter settings are provided in the Table 4. Table 3 provides the hyperparameters for fine-tuning with LoRA shared with other baselines. Our experiments on Llama2-7B were conducted using 2 NVIDIA RTX A5000 GPUs for parallel training, while experiments on RoBERTa-base were conducted using 4 NVIDIA RTX A5000 GPUs for parallel training.

## B.2 Evaluation Metrics for Uncertainty Estimation

Negative Log-Likelihood (**NLL**) and Expected Calibration Error (**ECE** [31]) are two prevalent metrics for assessing uncertainty estimation. NLL calculates the sum of the negative expected log probability of predicting the actual label. Suppose this predicted probability is given by the model $P_{\boldsymbol{\theta}}$, and we have a test dataset $\{\boldsymbol{x}_n, y_n\}_{n=1}^{N}$ of size $N$. Then the NLL measured on this dataset is

$$\text{NLL} = \tfrac{1}{N} \sum_{n=1}^{N} -\log P_{\boldsymbol{\theta}}(y_n). \tag{73}$$

This metric prefers models that assign higher probabilities to correct labels. If the model exhibits over-confident in an incorrect prediction, the probability assigned to the correct label will be diminished, thereby increasing the NLL.

On the other hand, ECE measures how well the model's confidence matches its accuracy. This is done by binning the predictions based on their confidence levels and then computing a weighted average of the absolute difference between accuracy and confidence within each bin:

$$\text{ECE} = \sum_{m=1}^{M} \tfrac{|B_m|}{n} \left| \text{acc}(B_m) - \text{conf}(B_m) \right|, \tag{74}$$

where $\text{acc}(B_m)$ and $\text{conf}(B_m)$ denote the average accuracy and confidence within bin $B_m$, respectively. These are given by:

$$\text{acc}(B_m) = \tfrac{1}{|B_m|} \sum_{i \in B_m} \mathbf{1}(\widehat{y}_i = y_i), \quad \text{conf}(B_m) = \tfrac{1}{|B_m|} \sum_{i \in B_m} P(\widehat{y}_i), \tag{75}$$

where $|B_m|$ is the number of samples in bin $m$. We set $|B_m| = 15$ across all experiments.

## B.3 Dataset Details

Table 5 summarizes the size of the training set and the number of labels for each dataset. Table 6 summarizes the prompt templates used for common sense reasoning tasks.

Table 5: Size of the training set and number of labels for each dataset.

| | WG-S [82] | ARC-C [18] | ARC-E [18] | WG-M [82] | OBQA [65] | BoolQ [17] | RTE [19] | MRPC [24] | WiC [76] | CoLA [104] |
|---|---|---|---|---|---|---|---|---|---|---|
| Size of Train. Set | 640 | 1.12k | 2.25k | 2.56k | 4.96k | 2.49k | 3.67k | 5.43k | 8.55k | 9.43k |
| Num. of Labels | 2 | 5 | 5 | 2 | 4 | 2 | 2 | 2 | 2 | 2 |

Table 6: Prompt templates for common sense reasoning tasks.

| Task | Prompt |
|---|---|
| Winogrande (WG-S/WG-M) | Select one of the choices that answers the following question: {question} Choices: A. {option1}. B. {option2}. Answer: |
| ARC (ARC-C/ARC-E), Openbook QA (OBQA), MMLU | Select one of the choices that answers the following question: {question} Choices: A. {choice1}. B. {choice2}. C. {choice3}. D. {choice4}. Answer: |
| BoolQ | Answer the question with only True or False: {question} Context: {passage}. |

## C  Additional Experimental Results

This section provides additional experimental results omitted from the main body of the paper due to space limitations. First, we present the results of BLoB when applied to RoBERTa, another pre-trained language model, in Appendix C.1. Next, in Appendix C.2, we conduct the ablation

Table 7: **Performance of different methods applied to LoRA on RoBERTa-base pre-trained weights.** The evaluation is undertaken on five GLUE [96] and SuperGLUE [95] tasks, with a shared hyper-parameter setting without using individual validation dataset. "↑" and "↓" represent that higher and lower values are preferred, respectively. The **boldface** and underline are used to denote the best and runner-up performance, respectively. The asterisk "*" denotes training failure.

| Metric | Method | Datasets | | | | |
|---|---|---|---|---|---|---|
| | | RTE [19] | MRPC [24] | WiC [76] | CoLA [104] | BoolQ [17] |
| ACC (↑) | MLE | $75.81_{\pm0.78}$ | $86.27_{\pm0.69}$ | $64.52_{\pm0.91}$ | $83.29_{\pm0.16}$ | $77.67_{\pm0.51}$ |
| | MAP | $75.81_{\pm2.26}$ | $86.36_{\pm0.51}$ | $65.46_{\pm1.04}$ | $83.00_{\pm0.15}$ | $77.69_{\pm0.65}$ |
| | MCD [29] | $76.65_{\pm0.85}$ | $87.75_{\pm0.53}$ | $68.55_{\pm0.32}$ | $84.76_{\pm0.62}$ | $78.41_{\pm0.25}$ |
| | ENS [51, 8, 103] | $77.74_{\pm1.10}$ | $88.64_{\pm0.37}$ | $65.83_{\pm0.41}$ | $84.08_{\pm0.44}$ | $78.57_{\pm0.36}$ |
| | BBB [11] | $49.46^*_{\pm2.53}$ | $68.38_{\pm0.00}$ | $50.57^*_{\pm1.74}$ | $69.13_{\pm0.00}$ | $62.16_{\pm0.04}$ |
| | LAP [116] | $76.05_{\pm0.95}$ | $86.52_{\pm0.72}$ | $64.52_{\pm0.91}$ | $83.29_{\pm0.16}$ | $77.67_{\pm0.52}$ |
| | BLoB (N=0) | $76.05_{\pm0.17}$ | $88.24_{\pm0.00}$ | $63.17_{\pm0.22}$ | $80.92_{\pm0.70}$ | $74.80_{\pm2.10}$ |
| | BLoB (N=5) | $74.61_{\pm0.61}$ | $88.48_{\pm0.60}$ | $64.00_{\pm0.53}$ | $80.54_{\pm0.16}$ | $74.77_{\pm1.77}$ |
| | BLoB (N=10) | $75.45_{\pm0.51}$ | $88.73_{\pm0.35}$ | $64.26_{\pm1.00}$ | $80.89_{\pm0.24}$ | $75.49_{\pm1.60}$ |
| ECE (↓) | MLE | $20.59_{\pm1.25}$ | $11.13_{\pm1.05}$ | $25.72_{\pm0.83}$ | $10.70_{\pm0.49}$ | $10.02_{\pm0.71}$ |
| | MAP | $21.67_{\pm3.25}$ | $11.12_{\pm0.45}$ | $24.26_{\pm1.17}$ | $10.61_{\pm0.49}$ | $10.11_{\pm0.62}$ |
| | MCD [29] | $13.06_{\pm0.59}$ | $7.36_{\pm0.85}$ | $16.94_{\pm0.75}$ | $4.58_{\pm0.27}$ | $6.21_{\pm0.33}$ |
| | ENS [51, 8, 103] | $19.47_{\pm0.37}$ | $10.13_{\pm0.56}$ | $28.62_{\pm0.63}$ | $12.44_{\pm0.42}$ | $5.98_{\pm0.26}$ |
| | BBB [11] | $2.66^*_{\pm2.24}$ | $6.46_{\pm0.43}$ | $2.53^*_{\pm0.39}$ | $3.90^*_{\pm0.41}$ | $5.02^*_{\pm0.29}$ |
| | LAP [116] | $\mathbf{5.33}_{\pm\mathbf{0.60}}$ | $6.29_{\pm0.99}$ | $\mathbf{11.48}_{\pm\mathbf{0.67}}$ | $\mathbf{3.13}_{\pm\mathbf{0.28}}$ | $4.84_{\pm0.15}$ |
| | BLoB (N=0) | $14.64_{\pm0.75}$ | $5.61_{\pm0.06}$ | $18.93_{\pm1.39}$ | $10.90_{\pm0.24}$ | $5.80_{\pm0.41}$ |
| | BLoB (N=5) | $10.46_{\pm0.61}$ | $4.49_{\pm0.32}$ | $13.62_{\pm1.18}$ | $7.76_{\pm0.21}$ | $3.21_{\pm0.13}$ |
| | BLoB (N=10) | $8.97_{\pm0.98}$ | $\mathbf{3.30}_{\pm\mathbf{0.19}}$ | $13.03_{\pm0.85}$ | $7.83_{\pm0.27}$ | $\mathbf{2.90}_{\pm\mathbf{0.12}}$ |
| NLL (↓) | MLE | $1.11_{\pm0.02}$ | $0.62_{\pm0.02}$ | $1.19_{\pm0.03}$ | $0.53_{\pm0.02}$ | $0.56_{\pm0.01}$ |
| | MAP | $1.23_{\pm0.10}$ | $0.58_{\pm0.04}$ | $1.14_{\pm0.06}$ | $0.53_{\pm0.00}$ | $0.55_{\pm0.02}$ |
| | MCD [29] | $0.65_{\pm0.03}$ | $0.39_{\pm0.04}$ | $0.88_{\pm0.02}$ | $\mathbf{0.39}_{\pm\mathbf{0.01}}$ | $0.50_{\pm0.01}$ |
| | ENS [51, 8, 103] | $1.04_{\pm0.05}$ | $0.63_{\pm0.02}$ | $1.70_{\pm0.07}$ | $0.62_{\pm0.00}$ | $\mathbf{0.48}_{\pm\mathbf{0.00}}$ |
| | BBB [11] | $0.69^*_{\pm0.00}$ | $0.63_{\pm0.00}$ | $0.69^*_{\pm0.00}$ | $0.62_{\pm0.00}$ | $0.67_{\pm0.00}$ |
| | LAP [116] | $0.55_{\pm0.00}$ | $0.47_{\pm0.01}$ | $\mathbf{0.63}_{\pm\mathbf{0.00}}$ | $0.48_{\pm0.00}$ | $0.53_{\pm0.00}$ |
| | BLoB (N=0) | $0.56_{\pm0.01}$ | $0.29_{\pm0.00}$ | $0.76_{\pm0.02}$ | $0.52_{\pm0.01}$ | $0.52_{\pm0.02}$ |
| | BLoB (N=5) | $0.50_{\pm0.01}$ | $0.27_{\pm0.00}$ | $0.68_{\pm0.01}$ | $0.45_{\pm0.01}$ | $0.51_{\pm0.02}$ |
| | BLoB (N=10) | $\mathbf{0.48}_{\pm\mathbf{0.01}}$ | $\mathbf{0.26}_{\pm\mathbf{0.00}}$ | $0.67_{\pm0.01}$ | $0.46_{\pm0.01}$ | $0.51_{\pm0.01}$ |

study on our proposed refinement in BLoB. Then we analyze the memory and training time costs in Appendix C.3. Finally, we provide visualization illustrating our BLoB's advantage on embedding uncertainty in Appendix C.6.

## C.1  Performance of RoBERTa on In-distribution Datasets

We also evaluate different methods on RoBERTa-base, which has approximately $1/50$ the parameter count of Llama2-7B. Table 7 shows the results. Compared to MLE, MAP shows minor improvements in NLL and ECE, while MCD, ENS, and LAP enjoy significant improvements. The convergence difficulty observed with the BBB algorithm is further exacerbated on the smaller model, resulting in significant decreases in ACC across all datasets, and even training failures on RTE and WiC. In contrast, our method demonstrates the best or runner-up performance in uncertainty estimation on almost all datasets. Only a slight decrease in ACC is observed on BoolQ and CoLA. We suspect that such decrease is caused by RoBERTa-base's small model size compared to the large size of these datasets BoolQ and CoLA (i.e., underfitting). Using a larger pretrained model, e.g., Llama2-7B, would potentially address this issue.

## C.2  Ablation Study

We perform an ablation study on the Llama2-7B model to showcase the effects of a range of techniques we designed: KL Re-Weighting (RW, Appendix B.1), Re-Parameterization (RP, Sec. 3.3), and Asymmetric Bayesianization (AB, Sec. 3.1). In the scenarios w/o AB, we Bayesianize both matrices, $A$ and $B$. In practice, using identical initialization and prior for the standard deviation

Table 8: **Ablation study of BLoB, applied to LoRA on Llama2-7B pre-trained weights,** where **RW**, **RP**, and **AB** represent our designed techniques of KL **R**e-**W**eighting (Appendix B.1), **R**e-**P**arameterization (Sec. 3.3), and **A**symmetric **B**ayesianization (Sec. 3.1), respectively. The evaluation is done following Table 1. We set the number of samples during training $K = 1$ and the number of samples during inference $N = 10$ across the variants (denoted by "−") of the BBB [11] and BLoB for fair comparison. We denote by "*" experiments with the scaled standard deviation matrix. The hyphen "−" in the table denotes training failure caused by "NaN" loss. "↑" and "↓" indicate that higher and lower values are preferred, respectively. **Boldface** and underlining denote the best and the second-best performance, respectively.

| Metric | Method | Techniques | | | Datasets | | | | | |
| --- | --- | --- | --- | --- | --- | --- | --- | --- | --- | --- |
| | | RW | RP | AB | WG-S [82] | ARC-C [18] | ARC-E [18] | WG-M [82] | OBQA [65] | BoolQ [17] |
| ACC (↑) | MLE | - | - | - | $68.99_{\pm0.58}$ | $69.10_{\pm2.84}$ | $85.65_{\pm0.92}$ | $74.53_{\pm0.66}$ | $81.52_{\pm0.25}$ | $86.53_{\pm0.28}$ |
| | BBB- | | | | - | - | - | - | - | - |
| | BBB-* | | | | - | - | - | - | - | - |
| | BBB [11] | | | ✓ | $56.54_{\pm7.87}$ | $68.13_{\pm1.27}$ | $85.86_{\pm0.74}$ | $73.63_{\pm2.44}$ | **$82.06_{\pm0.59}$** | **$87.21_{\pm0.22}$** |
| | BLoB- | ✓ | | ✓ | **$69.75_{\pm0.60}$** | $67.91_{\pm1.43}$ | $86.03_{\pm0.74}$ | **$76.24_{\pm0.55}$** | $81.65_{\pm0.66}$ | **$87.23_{\pm0.42}$** |
| | BLoB- | | ✓ | ✓ | - | - | - | - | - | - |
| | BLoB- | ✓ | ✓ | | - | - | - | - | - | - |
| | BLoB-* | ✓ | ✓ | | **$69.75_{\pm0.26}$** | **$70.27_{\pm0.48}$** | **$86.33_{\pm0.44}$** | $74.92_{\pm0.19}$ | $81.32_{\pm0.41}$ | $86.47_{\pm0.46}$ |
| | BLoB (Ours) | ✓ | ✓ | ✓ | $69.07_{\pm0.34}$ | $68.81_{\pm1.09}$ | $85.56_{\pm0.35}$ | $73.69_{\pm0.17}$ | $81.52_{\pm0.74}$ | $86.99_{\pm0.24}$ |
| ECE (↓) | MLE | - | - | - | $29.83_{\pm0.58}$ | $29.00_{\pm1.97}$ | $13.12_{\pm1.39}$ | $20.62_{\pm0.74}$ | $12.55_{\pm0.46}$ | $3.18_{\pm0.09}$ |
| | BBB- | | | | - | - | - | - | - | - |
| | BBB-* | | | | - | - | - | - | - | - |
| | BBB [11] | | | ✓ | $21.81_{\pm12.95}$ | $26.23_{\pm1.47}$ | $12.28_{\pm0.58}$ | $15.76_{\pm4.71}$ | $11.38_{\pm1.07}$ | $3.74_{\pm0.10}$ |
| | BLoB- | ✓ | | ✓ | $26.60_{\pm0.78}$ | $26.24_{\pm0.94}$ | $11.53_{\pm0.52}$ | $18.05_{\pm0.76}$ | $12.36_{\pm0.42}$ | $3.05_{\pm0.09}$ |
| | BLoB- | | ✓ | ✓ | - | - | - | - | - | - |
| | BLoB- | ✓ | ✓ | | - | - | - | - | - | - |
| | BLoB-* | ✓ | ✓ | | $16.59_{\pm0.57}$ | $13.85_{\pm1.06}$ | $5.93_{\pm0.63}$ | $8.33_{\pm0.78}$ | $4.77_{\pm0.26}$ | **$1.18_{\pm0.20}$** |
| | BLoB (Ours) | ✓ | ✓ | ✓ | **$9.35_{\pm1.37}$** | **$9.59_{\pm1.88}$** | **$3.64_{\pm0.53}$** | **$3.01_{\pm0.12}$** | **$3.77_{\pm1.47}$** | $1.41_{\pm0.19}$ |
| NLL (↓) | MLE | - | - | - | $3.17_{\pm0.37}$ | $2.85_{\pm0.27}$ | $1.17_{\pm0.13}$ | $0.95_{\pm0.07}$ | $0.73_{\pm0.03}$ | $0.32_{\pm0.00}$ |
| | BBB- | | | | - | - | - | - | - | - |
| | BBB-* | | | | - | - | - | - | - | - |
| | BBB [11] | | | ✓ | $1.40_{\pm0.55}$ | $2.23_{\pm0.04}$ | $0.91_{\pm0.06}$ | $0.84_{\pm0.15}$ | $0.66_{\pm0.05}$ | **$0.31_{\pm0.00}$** |
| | BLoB- | ✓ | | ✓ | $1.96_{\pm0.20}$ | $2.31_{\pm0.13}$ | $0.84_{\pm0.03}$ | $0.87_{\pm0.01}$ | $0.68_{\pm0.00}$ | **$0.31_{\pm0.00}$** |
| | BLoB- | | ✓ | ✓ | - | - | - | - | - | - |
| | BLoB- | ✓ | ✓ | | - | - | - | - | - | - |
| | BLoB-* | ✓ | ✓ | | $0.80_{\pm0.02}$ | $0.91_{\pm0.04}$ | $0.46_{\pm0.01}$ | $0.55_{\pm0.01}$ | $0.51_{\pm0.00}$ | $0.32_{\pm0.00}$ |
| | BLoB (Ours) | ✓ | ✓ | ✓ | **$0.63_{\pm0.01}$** | **$0.78_{\pm0.02}$** | **$0.40_{\pm0.01}$** | **$0.54_{\pm0.00}$** | **$0.50_{\pm0.01}$** | **$0.31_{\pm0.00}$** |

matrix $G$ of the variational distribution on both $A$ and $B$ leads to training failures caused by "NaN" loss across all datasets; this is consistent with the findings in Sec. 3.1. As a solution, we introduce a scaled standard deviation matrix $G/100$ on $B$ to alleviate early-stage fluctuations. Nevertheless, it is important to note that not using AB incurs double the additional memory cost and training time, as described in Appendix C.3.

As demonstrated in Table 8, BBB w/o AB fails to converge due to the unbounded NaN loss, which cannot be solved by using scaled standard deviation. By introducing KL Re-Weighting, Re-Parameterization, and scaled standard deviation, BLoB w/o AB achieves the runner-up performance and improves accuracy on small datasets. However, BLoB with all techniques achieves the best ECE and NLL with minimal additional computational cost.

## C.3 Additional Results on Memory and Time Efficiency

By introducing an additional standard deviation matrix $\Omega$ of the same size as the LoRA $A$ matrix, the number of trainable parameters in BLoB increases by half compared to LoRA. In the case of BLoB w/o Asymmetric Bayesianization (AB), the number of trainable parameters are twice as many as those in LoRA. The calculation of KL divergence and the inclusion of the additional standard deviation matrix in the likelihood loss computation result in additional forward and backward propagation time. We conduct parallel training using two NVIDIA RTX A5000 GPUs to observe the differences in GPU memory cost and training time between BLoB and standard LoRA fine-tuning on the Llama2-7B

Table 9: **A comparison of time and maximum memory cost between standard LoRA and BLoB, during training.** The evaluation is based on fine-tuning for 5,000 steps on the Llama2-7B model.

| Metric | Method | Datasets | | | | | |
|---|---|---|---|---|---|---|---|
| | | WG-S [82] | ARC-C [18] | ARC-E [18] | WG-M [82] | OBQA [65] | BoolQ [17] |
| Time (Seconds) (↓) | Standard LoRA | 1399 | 1614 | 1586 | 1408 | 1822 | 3382 |
| | BLoB (N=10) | 1563 | 1790 | 1753 | 1556 | 2142 | 3733 |
| Max Memory (MB) (↓) | Standard LoRA | 14688 | 16870 | 17044 | 14710 | 14984 | 20784 |
| | BLoB (N=10) | 15015 | 18863 | 19015 | 15015 | 15890 | 23552 |

Table 10: **A comparison of time and max memory cost between Standard LoRA, LAP, and BLoB (N=10) during inference.**

| Metric | Method | Datasets | | | | | |
|---|---|---|---|---|---|---|---|
| | | WG-S [82] | ARC-C [18] | ARC-E [18] | WG-M [82] | OBQA [65] | BoolQ [17] |
| Time (Seconds) (↓) | Standard LoRA | 17 | 5 | 8 | 17 | 7 | 58 |
| | LAP | 311 | 445 | 814 | 554 | 1165 | 2508 |
| | BLoB (N=10) | 193 | 45 | 86 | 193 | 75 | 627 |
| Max Memory (MB) (↓) | Standard LoRA | 14391 | 14157 | 13081 | 14391 | 14391 | 14160 |
| | LAP | 43742 | 61881 | 64737 | 43678 | 55642 | 67364 |
| | BLoB (N=10) | 14171 | 14411 | 13911 | 14407 | 14478 | 14177 |

model. The results are shown in Table 9. BLoB increases memory cost by only about 3% to 13% compared to LoRA, with training time increased by about 15%.

We further evaluate the inference time and maximum memory usage for standard LoRA, Laplace-LoRA (LAP), and our BLoB, as shown in Table 10. The experiments are conducted on two NVIDIA A100 GPUs. These results show that compared to LAP, our BLoB can achieve comparable or better ECE and NLL with less inference time and less memory usage. Notably, our BLoB's memory overhead compared to standard LoRA is minimal, while LAP introduces significant memory overhead.

## C.4 Impact of Sample Size on Inference Performance

Here we provide a more detailed empirical study on the sample size of BLoB during inference. Specifically, we report the results for different number of samples from $N = 1$ to $N = 160$ on the WG-S dataset, demonstrating improved uncertainty estimation with increased number of samples, as shown in Fig. 3.

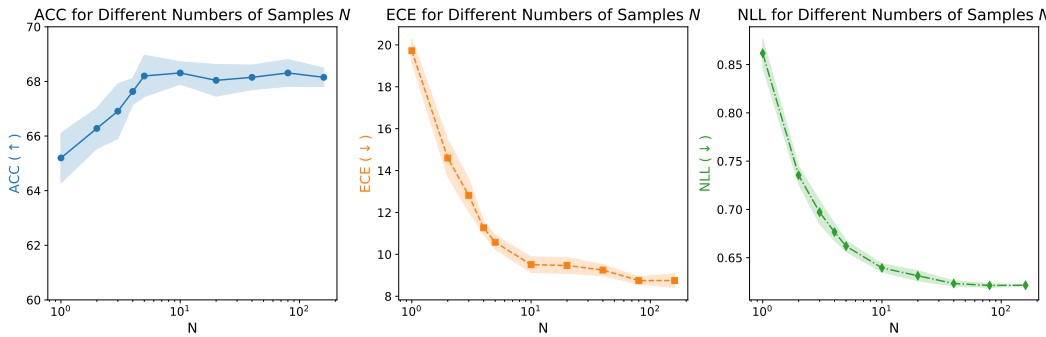

Figure 3: **Performance of BLoB with Varying Sample Sizes $N$ during Inference.** We fine-tune the Llama2-7B model on the WG-S dataset for 5,000 steps, evaluating the model's performance with different sample sizes, specifically when $N$ is 1, 2, 3, 4, 5, 10, 20, 40, 80, and 160.

## C.5 Trade-Off between Accuracy and Calibration Controlled by Gaussian Prior

The empirical trade-off between accuracy and calibration caused by different model architectures is observed in [85]. By controlling the standard deviation of the prior Gaussian distribution, we

observed a similar trade-off between accuracy and calibration. Specifically, we report results for different prior Gaussian standard deviations, ranging from 0.05 to 0.25, while proportionally scaling the learning rate of KL divergence from its original value of 0.01 to values between 0.0025 and 0.0125. This highlights the trade-off between accuracy and calibration, as shown in Fig. 4.

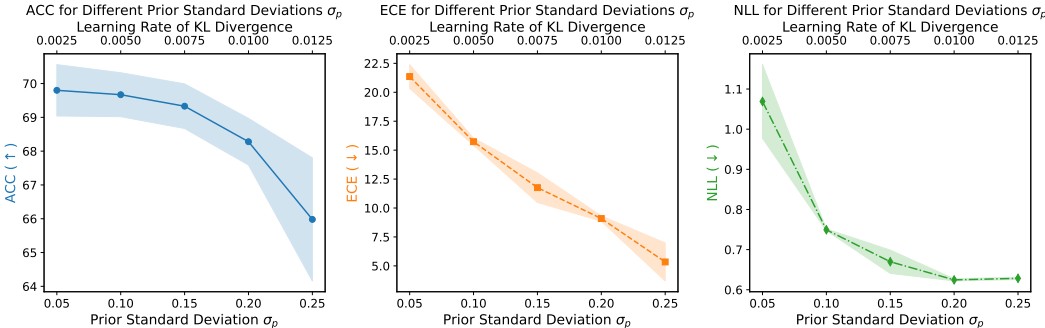

Figure 4: **Performance of BLoB (N=10) with Varying Prior Gaussian Standard Deviations** $\sigma_p$. We fine-tune the Llama2-7B model on the WG-S dataset for 5,000 gradient steps, evaluating the model's performance with different prior Gaussian standard deviations and learning rates of KL divergence.

## C.6   Embedding Uncertainty of BLoB: A Preliminary Visual Study

Estimating the uncertainty of LLMs in the embedding space has recently garnered significant attention in the community [13]. Expressing models' uncertainty via their generated embeddings can benefit both discriminative (the focus of this paper) and generative models. In this section, we present a preliminary study on uncertainty estimation in the embedding spaces of different models, as illustrated in Fig. 5. We compare BLoB with two baseline models, BBB and MCD, which can generate embedding samples and effectively estimate uncertainty. We exclude LAP from this section due to its excessive memory consumption, which consistently results in Out-Of-Memory (OOM) errors during inference. The experiment is conducted on the OBQA dataset [65], which consists of four categories.

For each input sequence $s$, we use the last token's embedding generated by the final transformer block in Llama2-7B as the final embedding. Given the weights $W$, we denote the embedding as $\phi(s; W)$. Generally, three types of embeddings can be generated using the Bayesian approach:

(a) Embeddings generated by the mean of the weights (these embeddings are shown as "★" in Fig. 5):

$$\phi(s; \mathbb{E}_{W \sim q(\cdot|\theta)}[W]) = \phi(s; W_0 + BM). \tag{76}$$

(b) Embedding samples generated by sampling different weights from the approximate posterior, whose distribution is plotted by the solid line (——).

(c) The expectation of the embedding, which is approximated by averaging the sampled embeddings:

$$\mathbb{E}_{W \sim q(\cdot|\theta)}[\phi(s; W)] \approx \frac{1}{N} \sum_{n=1}^{N} \phi(s; W_0 + B(M + E_n \circ \Omega)), \tag{77}$$

where $N$ denotes the number of samples during inference, and $E_n$ denotes the $n$-th sampled noise for the weight matrix. We show this expectation as "▼" in Fig. 5.

To visually demonstrate the confidence calibration effect of the Bayesian treatment, we adopt the following pipeline of visualization, which we believe can be further applied in visualizing other frameworks' embedding uncertainty quality.

(1) Acquire high-dimensional embeddings produced by the weight mean for the given test dataset, as decribed in (a) and Eqn. 76 above.

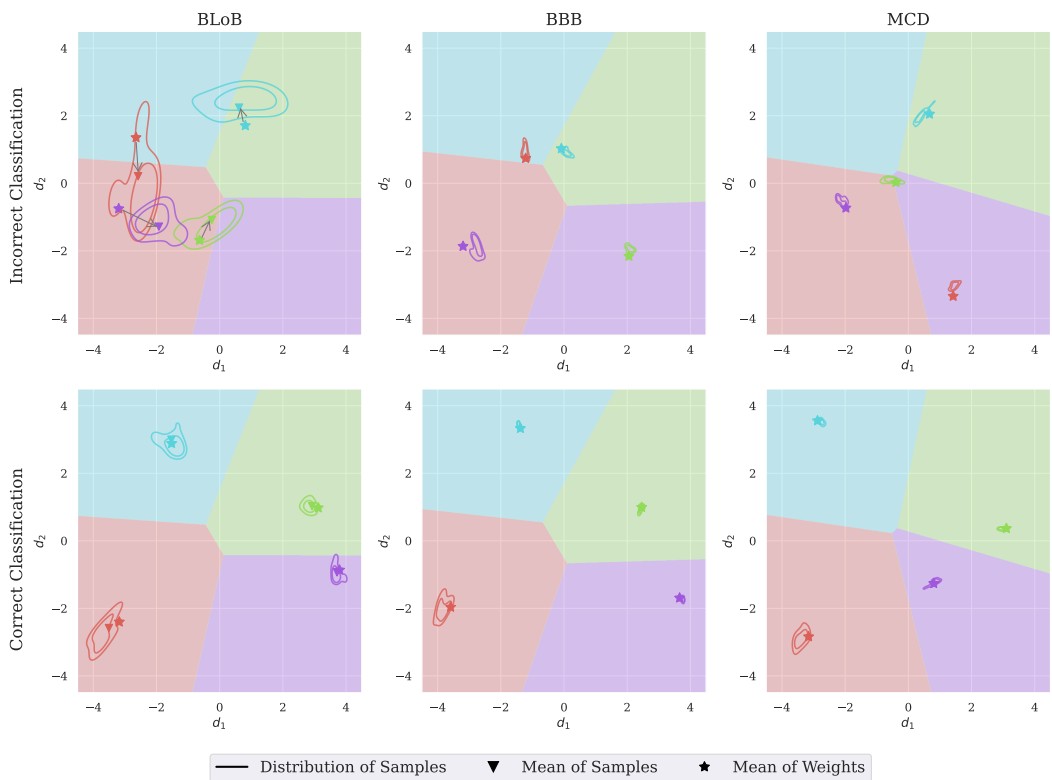

Figure 5: **Visualization of embedding uncertainty quality for different methods.** The model is fine-tuned for 5,000 steps on the Llama2-7B. We fine-tune the Llama2-7B model on the OBQA dataset for 5000 steps. The two contour lines represent the probability mass of 0.5 and 0.75, respectively.

(2) Use Linear Discriminant Analysis (LDA) [10] to project these high-dimensional embeddings into a low-dimensional 2D space.

(3) In the 2D space, fit a logistic regression model to mimic the decision regions and color them based on the true labels.

(4) Sample weights 10 times from the approximate posterior, generate the embeddings, and project them into the same 2D space using the previously learned LDA. Use Kernel Density Estimation (KDE) [79, 74] to show their distributions, as described in (b) above.

(5) Average the sampled embeddings for each example and visualize them in the 2D space, as described in (c) and Eqn. 77 above.

In Fig. 5, we show 4 correct and incorrect predictions made by each model. Ideally, a model with better uncertainty estimation should produce lower level of uncertainty (**smaller embedding variance**, i.e., smaller contours, and **further away from the decision boundary**) for correct predictions, and higher level of uncertainty (**larger embedding variance**, i.e., larger contours, and **closer to the decision boundary**). From the figure, we have the following observations:

- All three Bayesian approaches produce higher embedding variance for incorrect predictions and lower embedding variance for correct predictions. However, BLoB achieves significantly larger embedding variance compared to the baselines, consistent with the quantitative evaluation shown in Table 1. BLoB's produced variance is higher for the incorrect predictions, demonstrating its accurate uncertainty estimation even in the embedding space.

- In BLoB, the mean embedding produced by sampling weights from the approximate posterior is closer to the decision boundary than the embedding generated by the mean of weights (★→▼). This effect is most apparent when the prediction is incorrect, consistent with the quantitative results yielded from the final softmax layer of the model. Again, this demonstrates BLoB's Bayesian inference can bring the final prediction closer to the ground truth.

