# OpenReview forum: "BLoB: Bayesian Low-Rank Adaptation by Backpropagation for Large Language Models"
_NeurIPS.cc/2024/Conference — NeurIPS 2024 poster_

### Official Review · Reviewer_AGd6 · 2024-07-08

**Soundness:** 4
**Presentation:** 4
**Contribution:** 3
**Rating:** 7
**Confidence:** 3

**Summary:**

The paper proposes an algorithm to learn a (low-rank) variational posterior distribution over (a subset of) LLM weights during fine-tuning by combining Low-Rank Adaptation (LoRA) and Bayes by Backprop (BBB). To make the algorithm work in practice, several nontrivial modifications are introduced, such as certain parameterisations and optimisation strategies. Finally, the proposed method is empirically evaluated on a variety of in-distribution and out-of-distribution benchmarks and compared to relevant baselines.

**Strengths:**

- The paper is well written and the proposed algorithm is contextualised into existing work, with Figure 1 providing a great visual overview.
- Concepts are introduced in an adequate pace and order, and a clean notation makes it easy for the reader to follow.
- Elaborate empirical evaluation seems to suggest (somewhat) consistent gains over baseline approaches.
- While the proposed algorithm seems like a straightforward combination of LoRA and BBB, the paper mentions that certain modifications are necessary for the algorithm to work in practice. These modifications are a core part of the paper's contribution, and they are discussed thoroughly and justified with a mix of theoretical and empirical arguments.

**Weaknesses:**

- The proposed method requires additional compute time and memory compared to standard LoRA. While this information is provided in Appendix C.2 (and seems to be small), I encourage the authors to be transparent about this by briefly mentioning it in the main paper (if that is not already the case, perhaps I missed it).

**Questions:**

- Appendix B.1 mentions weights for the KL divergence term. How important is this? Please correct me if I am wrong, but I believe this was not explicitly discussed in the main paper. If this is important for the overall performance of the algorithm (and typically these kind of adjustments tend to be...), I suggest to at least mention it in the main paper (even if details remain in the appendix).
- Is there any particular reason why $\mathbf{A}$ is modelled in a probabilistic way instead of $\mathbf{B}$? In other words, would it also be possible to do the reverse, i.e. assume a prior over $\mathbf{B}$ and treat $\mathbf{A}$ deterministically? The choice seems arbitrary to me, but perhaps there is an underlying reason to it.

**Limitations:**

Theoretical assumptions are stated as part of the provided theorems. As far as I am aware, general limitations of the proposed method (beyond the ones which are being addresses by the proposed method itself) are not explicitly discussed and could at least be briefly mentioned in the conclusion.

---

> ### Author Rebuttal · Authors · 2024-08-05
>
> Thank you for the constructive and encouraging comments as well as the insightful questions. We are glad that you find our method ``"justified"`` by ``"theoretical and empirical arguments"``, our paper ``"well written"``/``"easy for the reader to follow"``, and the empirical evaluation showing our method's ``"consistent gains over baseline approaches"``. Additionally, we appreciate your recognition that our proposed necessary modifications are ``"a core part of the paper's contribution"``. Below we will address your questions in detail.
>
> **W1. The proposed method requires additional compute time and memory... by briefly mentioning it in the main paper...**
>
> This is a good suggestion. Following to your and Reviewer Adn3's suggestion, we conducted an additional study on the post-training computational cost for different methods. We calculate the inference time and maximum memory usage for LAP, standard LoRA, and BLoB. The results are presented in the table below:
>
> | Metric          | Method              | WG-S            | ARC-C           | ARC-E           | WG-M            | OBQA            | BoolQ           |
> | --------------- | ------------------- | --------------- | --------------- | --------------- | --------------- | --------------- | --------------- |
> |   | **Standard LoRA**       | 17              | 5               | 8               | 17              | 7               | 58              |
> |      **Inference Time (Seconds)**           | **LAP**                 | 311             | 445             | 814             | 554             | 1165            | 2508            |
> |                 | **BLoB (N=10)**         | 193             | 45              | 86              | 193             | 75              | 627             |
> ||
> |  | **Standard LoRA**       | 14391           | 14157           | 13081           | 14391           | 14391           | 14160           |
> |        **Max Memory (MB)**         | **LAP**                 | 43742           | 61881           | 64737           | 43678           | 55642           | 67364           |
> |                 | **BLoB (N=10)**         | 14171           | 14411           | 13911           | 14407           | 14478           | 14177           |
> ||
> |              | **Standard LoRA (MLE)** | 29.83 (0.58)    | 29.00 (1.97)    | 13.12 (1.39)    | 20.62 (0.74)    | 12.55 (0.46)    | 3.18 (0.09)     |
> |      **ECE**           | **LAP**                 | **4.15 (1.12)** | 16.25 (2.61)    | 33.29 (0.57)    | 7.40 (0.27)     | 8.70 (1.77)     | **1.30 (0.33)** |
> |                 | **BLoB (N=10)**         | 9.35 (1.37)     | **9.59 (1.88)** | **3.64 (0.53)** | **3.01 (0.12)** | **3.77 (1.47)** | 1.41 (0.19)     |
> ||
> |             | **Standard LoRA (MLE)** | 3.17 (0.37)     | 2.85 (0.27)     | 1.17 (0.13)     | 0.95 (0.07)     | 0.73 (0.03)     | 0.32 (0.00)     |
> |      **NLL**            | **LAP**                 | **0.63 (0.00)** | 1.03 (0.04)     | 1.38 (0.01)     | 0.57 (0.01)     | 1.00 (0.00)     | 0.45 (0.00)     |
> |                 | **BLoB (N=10)**         | **0.63 (0.01)** | **0.78 (0.02)** | **0.40 (0.01)** | **0.54 (0.00)** | **0.50 (0.01)** | **0.31 (0.00)** |
>
> These results show that compared to LAP, our BLoB can achieve comparable or better ECE and NLL with **less inference time** and **less memory usage**. Notably, our BLoB's memory overhead compared to standard LoRA is minimal, while LAP introduces significant memory overhead.
>
> We will include the discussion above in the main body of the revised paper as suggested.
>
> **Q1. ...the KL divergence term. How important is this? ... this was not explicitly discussed in the main paper...**
>
> This is a good question. Indeed, the KL divergence reweighting is crucial to BLoB's final performance. Due to space limit, we initially relegated this discussion to the Appendix. In our revision, we will present a comprehensive ablation study covering our novel KL reweighting schedule, parameterization, and asymmetric Bayesianization of LoRA, as suggested.
>
> **Q2. Is there any particular reason why $A$ is modelled in a probabilistic way instead of $B$?...**
>
> Thank you for this insightful question. As discussed in Section 3.1 (lines 129-131), BLoB models $A$ probabilistically rather than $B$ due to the conventional LoRA initialization: $A\leftarrow \text{random init}$, $B\leftarrow 0$. The choice is indeed arbitrary; this asymmetry could be reversed (i.e., model $B$ probabilistically instead) if the LoRA initialization were flipped (i.e., $A\leftarrow 0$, $B\leftarrow \text{random init}$).
>
> **Q3. Limitations**
>
> Please refer to **Q1. [Discussion of Limitations]** in **General Response**.

---

> > ### Comment · Reviewer_AGd6 · 2024-08-10
> >
> > Thank you for your response and for providing concrete numbers for inference time and memory consumption. I encourage you to include those in your manuscript. I am also glad to hear that you will present an ablation study about your KL reweighing schedule, as this is "crucial to BLoB's final performance". I am maintaining my score as I believe it is appropriate.

---

> > > ### Author Response · Authors · 2024-08-10
> > > **Thank you**
> > >
> > > Thank you very much for the encouraging comments and for acknowledging our contribution. We are glad that our response has been helpful and convincing. We will include the discussion and ablation study in the revision based on your suggestion.

---

### Official Review · Reviewer_Adn3 · 2024-07-08

**Soundness:** 2
**Presentation:** 2
**Contribution:** 2
**Rating:** 5
**Confidence:** 4

**Summary:**

This paper proposes a method called Blob, which uses mean-field variational inference on LoRA parameters to obtain Bayesian estimation on LLM. This will result in a richer posterior structure on the weight than diagonal, while maintaining the same computational cost as diagonal covariance. During training, a new parameterization is used for posterior covariance to accelerate convergence and Flipout is used to make training more stable. Empirical results show Blob obtains competitive performance compared with other Bayesian methods.

**Strengths:**

- By putting a diagonal prior and approximate posterior on LoRA parameters, it leads to a richer posterior covariance structure while maintaining the computational cost of diagonal covariance.

- The new parameterization of standard deviation seems promising for all VI training.

- Good performance compared with other Bayesian methods.

**Weaknesses:**

- Not clear about the method's limitations. Since it uses VI, common disadvantages of VI also apply: (1) it needs to do sampling when making predictions, which can be computationally expensive given the size of LLM. (2) Given the complex posterior structure, there will be a very high chance of getting noise samples, which leads to suboptimal performance. This is already demonstrated in Table 1 that using MC estimation decreases the ACC compared with only using the mean. In additional, it seems in practice many extra efforts are needed to get the method to work, e.g., the complex KL re-weighting and rescale size of training data mentioned in Appendix B.1.

- The whole paper feels a bit overselling: (1) I don't see the need to make Theorem 3.1 and Theorem 3.2 theorems, nothing surprising is stated there. (2) In Line 318, the authors claim their method "demonstrates superior generalization and uncertainty estimation capabilities", I don't think the results justify the "superior" claim. (3) In essence, Blob is mean-field VI on LoRA parameters trained with better strategies to make it suffer less from VI's high variance. I have no problem at explaining methods in great detail, but the whole writing on page 4 and page 5 feels like the authors are trying to make it sound as fancy as possible. I like the idea of the paper, but I think it can be presented in a more honest way.

**Questions:**

Major:
- I'm not sure what it mean for the mean matrix of q(A) to be an output of a neural network (Line 186-187). Does this mean you are using another neural network to fit the mean of posterior?

- In Eq. (70), it seems different mini-batch has different KL re-weighting hyperparamers lambda? How sensitive is the training to the value of lambda?

- What is the difference between BBB and Blob? From my understanding, the only difference is how you do the training. For BBB you use original parameterization and for Blob you use new parameterization plus Flipout? Also how many samples did you use to evaluate BBB in Table 1?

- I would be interested to see a figure plot of model performance versus the number of MC samples. Specifically, the y-axis is accuracy/ECE/NLL, the x-axis is the number of samples used during inference. This helps the reader to understand the sufficient sampling size for inference.

- I would be interested to see Laplace added to Table 8 to show a fair comparison of the computation cost compared with Blob.

- Why is Laplace performing so poorly on ARC-C dataset? The original Bayesian LoRA paper it seems to perform much better on that dataset.

Minor:
- Line 236, MLE and MAP are not uncertainty estimation methods.
- I don't entirely agree with the claim made in Lines 46-47. The possible suboptimal estimation of Laplace approximation comes from (1) LA, which assumes we reach MAP estimation while in practice you never know if the complex NN has really converged to MAP; (2) setting prior precision. It doesn't come from the post-training procedures.
- the notation "n" refers to two different things, at Eq. (4) it's the number of parameters, and at Line 153 it's the dimension of the weight matrix.
- Bolding is missing on ACC in Table 7.

**Limitations:**

See weaknesses and questions above.

---

> ### Author Rebuttal · Authors · 2024-08-05
>
> Thank you for your constructive feedback and insightful questions. We are glad that you like our idea and find our method ``"promising"`` and its performance ``"good"``/``"competitive"``. Below we address your major questions (W2, W3, Q3, and Q5) in detail. The remaining questions will be answered in a separate **Official Comment** after this **Rebuttal** post.
>
> **W2. It seems in practice many extra efforts are needed to get the method to work, e.g., the complex KL re-weighting and rescale size of training data mentioned in Appendix B.1.**
>
> Thanks for mentioning this. Our KL re-weighting technique ensures the model fits the data well while converging to the prior distribution, akin to the BBB approach [1]. By rescaling the training data size, we maintain consistent hyperparameters across varying dataset sizes.
>
> As Reveiwer AGd6 noted, these adjustments are not limitations but key technical contributions (``"a core part of the paper's contribution"``) that enable BLoB to work effectively across diverse datasets.
>
> [1] Blundell, Charles, et al. "Weight uncertainty in neural network." International conference on machine learning. PMLR, 2015.
>
> **W3.1. I don't see the need to make Theorem 3.1 and Theorem 3.2 theorems, nothing surprising is stated there... I like the idea of the paper, but I think it can be presented in a more honest way.**
>
> This is a good question. Actually our Theorems 3.1 & 3.2 are necessary because they
> + shows that with a proper $\tilde{R}$, one can compute the KL divergence for the high-dimensional full weight vec($W$) simply by computing the KL divergence for $A$, which is much lower-dimension, more parameter-efficient, more memory-efficient, and faster, and
> + investigate important theoretical properties of our BLoB, offering insights into its underlying assumptions and advantages.
>
> We will include the clarification above in our revision as suggested.
>
> **W3.2. ...I don't think the results justify the "superior" claim.**
>
> For the performance of BLoB (N=10), in the ID experiments in Table 1, it achieves the best ECE and NLL performance on almost all datasets, with similar ACC compared to MLE and MAP. In the OOD experiments in Table 2, BLoB ($N=10$) achieves the best performance in 7 out of 12 metrics across the four datasets, while the second-best LAP method achieves the best performance in only 4 metrics. The results clearly demonstrate that BLoB outperforms other baseline methods.
>
> **Q3. What is the difference between BBB and BLoB?... How many samples are used to evaluate BBB**
>
> The key distinctions between BLoB and BBB are:
> 1. **Asymmetric Bayesianization (AB):** BLoB models the approximate variational distributions for only one LoRA component $A$. This technique is crucial, as classic BBB consistently fails on various datasets without it. In fact, to produce meaningful results for BBB in Tables 1 and 2, BBB has to incorporate our proposed AB, giving it the baseline BBB an unfair advantage.
> 2. **Novel Parameterization and Training:** BLoB introduces a new method to parameterize the standard deviation of matrix $A$ and employs a different KL-reweighting strategy during training. Table 1 demonstrates that these innovations are essential for successful uncertainty estimation.
>
> As mentioned by Reviewer AGd6, these certain distinctions ``"are necessary for the algorithm to work in practice"`` and ``"are a core part of the paper's contribution, and they are discussed thoroughly and justified with a mix of theoretical and empirical arguments."``
>
> **Number of Samples.** For fair comparison, both BLoB and BBB use $N=10$ samples in Table 1. We will clarify this in the revision.
>
> **Q5. I would be interested to see Laplace added to Table 8 to show a fair comparison of the computation cost compared with Blob.**
>
> This is a good question. We divide the computational cost into two phases: training and post-training (inference on test data).
> LAP's training cost is equivalent to LoRA's, as shown in Table 8, since it only involves MAP training. For a comprehensive comparison of post-training computational costs, we calculate the inference time and maximum memory usage for LAP, standard LoRA, and BLoB. The results are presented in the table below. The complete table, including the corresponding ECE and NLL for reference, is provided as **Table 1** in the PDF attached to the **General Response**.
>
> | Metric          | Method        | WG-S            | ARC-C           | ARC-E           | WG-M            | OBQA            | BoolQ           |
> | --------------- | ------------- | --------------- | --------------- | --------------- | --------------- | --------------- | --------------- |
> | **Inference Time (Seconds)**  | **Standard LoRA** | 17              | 5               | 8               | 17              | 7               | 58              |
> |                 | **LAP**           | 311             | 445             | 814             | 554             | 1165            | 2508            |
> |                 | **BLoB (N=10)**   | 193             | 45              | 86              | 193             | 75              | 627             |
> | **Max Memory (MB)** | **Standard LoRA** | 14391           | 14157           | 13081           | 14391           | 14391           | 14160           |
> |                 | **LAP**           | 43742           | 61881           | 64737           | 43678           | 55642           | 67364           |
> |                 | **BLoB (N=10)**   | 14171           | 14411           | 13911           | 14407           | 14478           | 14177           |
>
> These results show that compared to LAP, our BLoB can achieve comparable or better performance with **less inference time** and **less memory usage**. Notably, our BLoB's memory overhead compared to standard LoRA is minimal, while LAP introduces significant memory overhead.
>
> **The remaining questions are answered in a separate Official Comment titled "Remaining Questions for Reviewer Adn3" below.**

---

> ### Author Response · Authors · 2024-08-07
> **Remaining Questions for Reviewer Adn3**
>
> **W1. Not clear about the method's limitations.**
>
> Please refer to **Q1. [Discussion of Limitations]** in **General Response**.
>
>
> **Q1. What it mean for the mean matrix of q(A) to be an output of a neural network (Line 186-187)?**
>
> We are sorry for the typo. $q(A)$ is not the output of a neural network. It is directly modeled as a Gaussian distribution $\mathcal{N}(M, \Omega)$, with $M$ and $\Omega$ as learnable parameters.
>
> **Q2. In Eq. (70), it seems different mini-batch has different KL re-weighting hyperparamers lambda? How sensitive is the training to the value of lambda?**
>
> Yes, different mini-batches has different $\lambda$ values. When we use the typical settings of $\lambda_i = \frac{2^{M-i}}{2^M-1}$ [1] or $\lambda_i = \frac{1}{M}$ [2], the model struggles to fit the data distribution while converging to the prior distribution, resulting in an "NaN" NLL loss. Our setting of $\lambda_i = \frac{2^i}{2^M-1}$ allows the model to fit to the data in the early stages of training and achieve a reasonable level of complexity cost in the later stages. This re-balance scheme of data likelihood and complexity cost has proven effective across multiple datasets and can be potentially applied to other VI methods.
>
> We will include the discussion above in the revision as suggested.
>
> [1] Blundell, Charles, et al. "Weight uncertainty in neural network." International conference on machine learning. PMLR 2015.
>
> [2] Graves, Alex. "Practical variational inference for neural networks." NeurIPS 2011.
>
>
> **Q4. I would be interested to see a figure plot of model performance versus the number of MC samples. ...**
>
> This is a good suggestion. Following your suggestion, we have run additional experiments for model performance versus the number of MC samples on WG-S dataset. The results are presented in the table below. The corresponding figure is provided as **Figure 1** in the PDF attached to the **General Response**.
>
> |                        | **N=0** | **N=1** | **N=2** | **N=3** | **N=4** | **N=5** | **N=10** | **N=20** | **N=40** | **N=80** | **N=160**  |
> | ---------------------- | ------- | ------- | ------- | ------- | ------- | ------- | -------- | -------- | -------- | -------- | ------ |
> | **ACC ($\uparrow$)**   | 71.44   | 65.20   | 66.28   | 66.91   | 67.63   | 68.20   | 68.31    | 68.04    | 68.15    | 68.31    | 68.15 |
> | **ECE ($\downarrow$)** | 19.71   | 19.72   | 14.60   | 12.82   | 11.27   | 10.58    | 9.51    | 9.47    | 9.25     | 8.75     | 8.75   |
> | **NLL ($\downarrow$)** | 0.84    | 0.8617  | 0.7355  | 0.6971  | 0.6766  | 0.6620  | 0.6395   | 0.6313   | 0.6233   | 0.6213   | 0.6215 |
>
> We can see that in general, the performance improves with more MC samples, but saturates after around $N=10$.
>
> **Q6. Why is Laplace performing so poorly on ARC-C dataset? The original Bayesian LoRA paper it seems to perform much better on that dataset.**
>
> Please refer to **Q2. [Reproducing LAP's Results on the ARC-C Dataset]** in **General Response**.
>
> **Q7. Line 236, MLE and MAP are not uncertainty estimation methods.**
>
> We apologize for the confusion. We will remove "uncertainty estimation" when describing MLE and MAP in the revision as suggested.
>
> **Q8. I don't entirely agree with the claim made in Lines 46-47. The possible suboptimal estimation of Laplace... never know if the complex NN has really converged to MAP; (2) setting prior precision...**
>
> This is a good point. We agree that these are also major reasons that lead to Laplace approximation's poor performance. We will include these two additional reasons in our revision as suggested.
>
> **Q9. Notation issue and missing Bolding**
>
> We are sorry for these typos and will correct them in the revision as suggested.

---

> > ### Comment · Reviewer_Adn3 · 2024-08-12
> >
> > Thank you for the rebuttal, it addresses most of my concerns. Still, I am not fully convinced that a Theorem is needed to state results in Theorem 3.1 and Theorem 3.2, maybe this is because I come from a more Bayesian background and thus the results do not look surprising.
> >
> > On a side note, I recently came across an ICML 2024 paper "Variational Learning is Effective for Large Deep Networks". In essence, they do natural gradient descent in natural parameter space for mean-field VI, and they make it work for large-scale neural networks including GPT2. This is highly relevant to the paper so I strongly suggest the author to discuss it in the paper.

---

> > > ### Author Response · Authors · 2024-08-13
> > > **Thank You for Your Further Feedback**
> > >
> > > Thank you for your continued feedback and for keeping the communication channel open. We are glad that you found our response helpful and that it addressed most of your concerns. Below we address your remaining comments in detail.
> > >
> > > **W3.1 [the Term "Theorem"]**
> > >
> > > We appreciate your insightful feedback. In light of your suggestion, we will revise Theorem 3.1 and Theorem 3.2 to Proposition 3.1 and Proposition 3.2, respectively. We believe this change can more accurately reflect the nature of our contributions. Nonetheless, we maintain that the discussion presented here offers valuable insights to the broader community. The theoretical foundation of Asymmetric Bayesianization laid out in this work can serve as a springboard for the development of new methods in the field.
> > >
> > > In addition, inspired by your comments, we plan to reorganize Section 3.1 to enhance clarity. We will start with the calculation of the full weight matrix $W\_{ij}=W\_{0,ij} + \sum\_{k=1}^r B\_{ik}A\_{kj}$ (i.e., Eqn. 5 of the paper) and subsequently introduce the advantages of Asymmetric Bayesianization over bayesianizing both $A$ and $B$.
> > >
> > > **[Related Work]**
> > >
> > > Thank you for bringing the ICML 2024 paper "Variational Learning is Effective for Large Deep Networks" to our attention. It was not available at the time of our submission. We agree that it is highly relevant to our work, and we will ensure it is cited and discussed in the revised version.
> > >
> > > Finally, we would like to express our sincere gratitude once again for your insightful and constructive comments, which have greatly helped improve our paper.

---

### Official Review · Reviewer_Zor8 · 2024-07-11

**Soundness:** 4
**Presentation:** 1
**Contribution:** 3
**Rating:** 6
**Confidence:** 5

**Summary:**

Proposes to have a Bayesian version of LoRA by placing priors over the low rank matrices. This evolves into placing prior over one of the low rank matrices. Inference is via variational inference. Results are mixed, though it shows the method is competitive with others.

**Strengths:**

- The eventual simplicity of the approach by placing prior over only one of the low rank matrices.
- Use of independent Gaussians in prior and posterior for efficiency.
- The method is competitive, based on the results.

**Weaknesses:**

The flow of the paper is rather poor, and the presentation of the method overly complicated. I prefer the authors to simply start from (5) and then say the deltas to the pre-trained weights are Gaussians because they are sum of Gaussians. The consideration to also put prior on $B$ can be deferred to discussions, and the trade-offs discussed there. This make the model presentation cleaner, and avoids the problem of having to involve improper priors in Theorem 3.1 (since B is low rank, $\Sigma_{q}$ is low rank, the Gaussian is degenerate) and Theorem 3.2, and the subsequent problem of reader having to think about how the KL with generate Gaussians would work out in theory. In short, the theorems are unnecessary and need fixing because of improper distributions.

1. Line 137: What is it that requires accurate estimation?
2. Section 4: How is more than one sample (for N>0) used in generating a single output for evaluation and then influence the evaluation metric?

*Minor*
- Eq (3) to remove the $\min_{\theta}$ prefix, and to introduce the prior and likelihood.
- Section 4: Shouldn't ECE and NLL be the metrics in focus because this is about "Bayesian"?
- Section 4: The numbers in the table does not justify saying BLoB is the best.
- "Bayesianization" is a mouthful --- can we think of a better derived word?

**Questions:**

Q1 and Q2 above (in weaknesses)

**Limitations:**

No. Paper need to include some discussion on the limitations of their approach/paper. I do not see any potential negative societal impact to be addressed.

---

> ### Author Rebuttal · Authors · 2024-08-05
>
> Thank you for your constructive feedback and insightful questions. We are glad that you find our proposed methodology ``"competitive"`` with ``"simplicity"``, ``"efficiency"``, and ``"good contribution"``. Below we will address your questions in detail.
>
> **W1. The flow of the paper is rather poor, and the presentation of the method overly complicated. I prefer the authors to simply start from (5) and...**
>
> We sincerely appreciate your valuable feedback on the organization of our paper. We will make every effort to incorporate your suggestions to ensure that our model presentation is clearer and more understandable.
>
> **W2. ...the problem of having to involve imprior priors... In short, the theorems are unneccessary and need fixing because of improper distributions.**
>
> This is a good question.
>
> **Degenerate Gaussians.** Note that in our context, **degenerate Gaussian distributions are still valid for probabilistic inference**, as demonstrated by Schoeman et al. [1] (Eq. 12, 16-18), because
> + their probability density is well-defined, and
> + the KL divergence between two degenerate Gaussians is computable.
>
> Due to space constraint, we provide detailed discussion in a separate  **Official Comment** titled **Degenerate Gaussians are Proper for Probabilistic Inference** below this **Rebuttal**.
>
> **Necessity of Theorems 3.1 & 3.2.** Our Theorems 3.1 & 3.2 are necessary because they
> + shows that with a proper $\tilde{R}$, one can compute the KL divergence for the high-dimensional full weight vec($W$) simply by computing the KL divergence for $A$, which is much lower-dimension, more parameter-efficient, more memory-efficient, and faster, and
> + investigate important theoretical properties of our BLoB, offering insights into its underlying assumptions and advantages.
>
> We will include the clarification above in our revision as suggested.
>
> [1] Schoeman, Johannes Cornelius, Corné E. van Daalen, and Johan A. du Preez. "Degenerate Gaussian factors for probabilistic inference." IJAR 2022.
>
> **W3. Eq (3) to remove the $\min_\theta$ prefix, and to introduce the prior and likelihood.**
>
> This is a good suggestion. We will remove the prefix in the revision.
>
> **W4. Section 4: Shouldn't ECE and NLL be the metrics in focus because this is about "Bayesian"?**
>
> Yes, ECE and NLL are indeed the primary metrics for evaluating LLMs' uncertainty estimation. However, we must also consider the model's accuracy to ensure a balance between uncertainty estimation and predictive performance. For example, a random classifier could achieve perfect calibration (zero/optimal ECE) and potentially lower NLL than a poorly calibrated model. Thus, while prioritizing NLL and ECE, it is also crucial to maintain ACC comparable to MLE and MAP baselines.
>
> **W5. Section 4: The numbers in the table does not justify saying BLoB is the best.**
>
> For the performance of BLoB ($N=10$), in the ID experiments in Table 1, it achieves the best ECE and NLL performance on almost all datasets, with similar ACC compared to MLE and MAP. In the OOD experiments in Table 2, BLoB ($N=10$) achieves the best performance in 7 out of 12 metrics across the four datasets, while the second-best LAP method achieves the best performance in only 4 metrics. The results clearly demonstrate that BLoB outperforms other baseline methods.
>
> **W6. "Bayesianization" is a mouthful --- can we think of a better derived word?**
>
> Thanks for mentioning this. We follow prior work to use "Bayesianization" as the noun form of "Bayesianize" [1]. This term accurately captures the process of transforming a deterministic component into a Bayesian one.
>
> [1] Bacharach, Michael, and Susan Hurley. "Issues and advances in the foundations of decision theory." Foundations of decision theory (1991): 1-38.
>
> **Q1. Line 137: What is it that requires accurate estimation?**
>
> The term "accurate estimation" refers to estimating $E_{A, B}[BAx]$. When both $A$ and $B$ are modeled as Gaussian posteriors, $E_{A, B}[BAx]=0$ holds before fine-tuning. However, accurately estimating this expectation requires an impractically large number of weight samples. In our BLoB approach, we model $B$ as deterministic and initialize it to 0 (similar to LoRA), avoiding this issue and leading to faster convergence with fewer samples.
> We will include the discussion above in the revision.
>
> **Q2. How does more than one sample (for N>0) used in generating a single output?**
>
> This is a good question. Ideally, a Bayesian neural network's output is the expected output $E_{q(W|\theta)}[P(Y|W,X)]$, where $X$ is the input, $Y$ is the output, and $q(W|\theta)$ is the approximate posterior of the parameters.
> In practice, we approximate this expectation through sampling: $E_{q(W|\theta)}[P(Y|W,X)]\approx \frac{1}{N}\sum_{n=1}^{N} P(Y|W_n, X)$, where $W_n\sim q(W|\theta)$ is the n-th sample drawn from the approximate posterior, and $N$ is the total number of samples.
>
> **Q3. Limitations**
>
> Please refer to **Q1. [Discussion of Limitations]** in **General Response**.

---

> ### Author Response · Authors · 2024-08-07
> **Degenerate Gaussians are Proper for Probabilistic Inference**
>
> In this separate Official Comment, we will elaborate on two key reasons why degenerate Gaussian distributions are valid for probabilistic inference:
>
> 1. **Their probability density is well-defined.**
>
> According to Schoeman et al. [1], the *probability density* of a degenerate Gaussian distribution can be factorized into the product of the density of a non-degenerate Gaussian distribution in a lower-dimensional linear subspace and a Dirac delta function (Eq. 12 in [1]). This density function has an alternative expression in the form of limit (Eq. 16-18 in [1]), which aligns with our main theorem 3.1 and 3.2:
> $$
> p(x) = \lim_{a\rightarrow 0}\mathcal{N}(x | \mu, Q(\Lambda^{-1} - aI)Q^T + a I) = \mathcal{N}(x | \mu, \Sigma=\lim_{a\rightarrow 0} Q(\Lambda^{-1} - aI)Q^T + a I).
> $$
> Here $\Lambda^{-1}$ is the precision matrix of the lower-dimensional non-degenerate Gaussian distribution, and $Q$ is the linear transformation matrix that expands this to high dimensions. In our main paper, we use the simplified notation $\mathcal{N}(x | \mu, \Sigma)$ for readability, omitting the limit. The full limit notation and proof appear in Appendix A.1. We will clarify this simplification in our revision.
>
> 2. **The KL divergence between two degenerate Gaussians is computable under certain conditions.**
>
> Secondly, the *KL divergence* between two degenerate Gaussians can be computed under the condition that they "have support on the same lower-dimensional manifold" (Eq. 44 in Schoeman et al. [1]). This condition has also been clearly stated as satisfied in our Theorem 3.2 ($RR^{T} = BB^{T}$, line 177), and we also include a detailed derivation of how we reach this condition in Appendix A.1 (line 676-678).
>
> [1] Schoeman, Johannes Cornelius, Corné E. van Daalen, and Johan A. du Preez. "Degenerate Gaussian factors for probabilistic inference." IJAR 2022.

---

> ### Comment · Reviewer_Zor8 · 2024-08-10
>
> W1 + W2. I understand that the method is sound because your degenerate Gaussians are on the same subspace. It is now clear to me that you also understand this constraint. I'll increase the soundness score, and the overall score.
>
> However, I still believe that the presentation can be vastly simplified (and hence the paper more accessible) by simply starting from (5). The relation to full Bayesianization can be deferred to discussion.
>
> W5.  It is "generally better" than the other methods. You can also say it is "the best on average" if you do some averaging either over the score or by counting wins. You cannot just say it is "the best".
>
> Q2. I see that the "experimental setup strictly adheres to the original LoRA [2] framework". It is good to say this explicitly in the paper. For ACC, do you generate, say N=10, and then count if *any* is correct? Is there some discount because you have N chances?

---

> > ### Author Response · Authors · 2024-08-10
> > **Thank You for Your Further Feedback**
> >
> > Thank you for your further feedback and for keeping the communication channel open. We are glad that you found our response helpful and that it addressed your concerns. Below we address your remaining comments in detail.
> >
> > **W1+W2 [Presentation]**: Thank you again for your suggestion regarding the presentation. We will re-organize the presentation to start with Eqn. 5 in the revision as you suggested.
> >
> > **W5 ["Generally Better"]**: Thank you for your suggestion. We will adjust the claim from "the best" to "generally better" to more accurately reflect our contribution.
> >
> > **Q2 [ACC and N=10 Samples]**: Thank you for your recognition and insightful question. For ACC, we first draw N=10 samples, i.e., N=10 output probabilities, from our BLoB, and then compute the average of these N samples as *one* single final prediction.
> >
> > This approach aligns with the method we previously described: "approximate this expectation through sampling: $E_{q(W|\theta)}[P(Y|W,X)] \approx \frac{1}{N} \sum_{n=1}^{N} P(Y|W_n, X)$." Based on this averaged output probability (i.e., the approximation of the expectation), we then calculate the ACC. The ACC is *not* calculated based on whether *any* individual sample is correct. Instead, N=10 samples are aggregated to make *only one prediction*. Therefore there is *no discount* needed.
> >
> > We will follow your suggestion to provide further clarification on the output of variational inference and clarify that "experimental setup strictly adheres to the original LoRA [2] framework" in our revised version.
> >
> > Last but not least, we would like to thank you again for your insightful and constructive comments. They have greatly help improve our paper.

---

> ### Author Response · Authors · 2024-08-12
> **Improved Outline of the Method**
>
> Thank you for your valuable suggestions on enhancing the presentation quality of our work. Following your suggestions, we have refined Sections 3.1 and 3.2. Below is the outline of our updated version:
>
> **Section 3.1: Low-Rank Variational Approximate Posterior Distribution**
>
> 1. Main Method: Introduction of Asymmetric Bayesianization Scheme
>    - Calculating each entry of the full weight matrix: $W\_{ij}=W\_{0,ij} + \sum\_{k=1}^r B\_{ik}A\_{kj}$ (i.e., Eqn. 5 of the paper)
>    - $A$ modeled by independent Gaussians: $P(A|\theta)=\mathcal{N}(A|M,\Omega^2)$
>    - $B$ modeled as deterministic values
>
> 2. Presentation of Theorem 3.1
>    - Statement: Asymmetric Bayesianization corresponds to assuming a low-rank Gaussian variational distribution for the full weight $W$
>
> 3. Discussion on the Choice of Asymmetric Bayesianization
>    - Advantages over Bayesianizing both $A$ and $B$:
>      + Stable training at the early stage
>      + Faster convergence of parameters during training
>      + Lower memory cost
>
> **Section 3.2: Low-Rank Prior Distribution**
>
> 1. Presentation of the Prior Distribution on the Low-Rank Component $A$
>
> 2. Corresponding Prior Distribution on the High-Dimensional Space of the Full Weight Matrix $W$
>
> 3. Presentation of Theorem 3.2
>    - Statement: Full-weight KL divergence can be efficiently computed on the low-rank component
>
> 4. Dedicated Remark on the Legitimacy of using Degenerate Gaussians for Probabilistic Inference (as in Our Previous Discussion)
>
> Thank you once again for your suggestions. We believe this revision will make the paper more accessible for our audience and help address any potential areas of confusion.

---

### Official Review · Reviewer_Lm4V · 2024-07-17

**Soundness:** 3
**Presentation:** 3
**Contribution:** 2
**Rating:** 6
**Confidence:** 3

**Summary:**

This work introduces a Bayesian Deep Learning framework for finetuning LLMs with probabilistic LoRA. Unlike existing work by Yang et al. [1] which uses a Laplace approximation, the variational distribution is parameterized using diagonal Gaussians as done in [2] (”Bayes by Backprop”).

Specifically, the authors design the variational distribution as follows:
Given the standard LoRA updates $W = W_0 + BA$, only $A$ is modeled probabilistically while $B$ is deterministic. In combination with an appropriate choice of prior, this allows for a a low-rank parameterization of the variational posterior and fast KL-computation in the lower dimensional parameter space.
Experiments compare the methods to existing baselines in terms of accuracy of the mean predictions, calibration (ECE) and NLL on a range of datasets.

[1] https://arxiv.org/pdf/2308.13111
[2] https://proceedings.mlr.press/v37/blundell15.pdf

____

I have updated my overall score +1 after the rebuttal.

**Strengths:**

The problem of assigning principled uncertainties to (fine-tuned) LLM outputs is important, and it makes sense to compare different Bayesian Deep Learning strategies in combination with LoRA. The exposition of the method and paper in general is clear.

**Weaknesses:**

Novelty is limited compared to Yang et al. [1], which implements a very similar idea via Laplace approximations instead of the Bayes-by-Backprop variational posterior. Additionally, the method by Yang et al. ("LAP") is not replicated successfully in Table 1. For example, for the LAP method on ARC-C datasets the results in this work fall behind the results in [1] by 50%:
- This work: Accuracy: 29.73±12.02, ECE: 14.24±1.65 and NLL: 1.53±0.01
- [1]: Accuracy: 66.91±1.1, ECE:  7.5±1.2 and NLL: 0.86±0.02, respectively.

This makes it difficult to gauge how much the approach improves compared to [1].

**Questions:**

Questions:
- Table 1: As $N \rightarrow \infty$, would we expect the accuracy to converge to those obtained with $N=0$ (mean prediction)? Is BLoB an unbiased estimator of the mean prediction?
- l. 240: “For MCD, we use an ensemble of 10 LoRAs with a dropout rate of p = 0.1”. Why is an ensemble used here? Shouldn’t the uncertainties come from the dropout variational posterior? Why are multiple models necessary?
- Will you open source the code on publication?

Suggestions & Typos:
- l. 33: I.e., unable to … (sentence is not grammatical, delete)
- l. 110: yielded by parameterization —> yielded by reparameterization
- Algorithm 1, line 11: Updater —> Update
- l. 237: Maximize A Posteriori —> Maximum A Posteriori
- l. 239: “For MAP, we use a weight decay rate of 1e-5.” I suggest to formulate this in terms of prior variance instead.
- l. 638 covaraince —> covariance

**Limitations:**

No, I recommend a more explicit treatment of the limitations, perhaps in a dedicated section.

---

> ### Author Rebuttal · Authors · 2024-08-05
>
> Thank you for your constructive feedback and insightful questions. We are glad that you find the problem we address ``"important"``, our empirical evaluation of BDL on LoRA ``"make sense"``, and our paper ``"clear"``. Below we will address your questions in detail.
>
> **W1. Novelty is limited compared to Yang et al. [1], which implements a very similar idea via Laplace approximations instead of the Bayes-by-Backprop variational posterior.**
>
> Please refer to **Q3. [Novelty of BLoB]** in **General Response**.
>
> **W2. The method by Yang et al. ("LAP") is not replicated successfully in Table 1.**
>
> Please refer to **Q2. [Reproducing LAP's Results on the ARC-C Dataset]** in **General Response**.
>
> **Q1. As $N\rightarrow \infty$, would we expect the accuracy to converge to those obtained with $N=0$ (mean prediction)? Is BLoB an unbiased estimator of the mean prediction?**
>
> This is a good question. Ideally, a Bayesian neural network's output is the expected output $E_{q(W|\theta)}[P(Y|W,X)]$, where $X$ is the input, $Y$ is the output, and $q(W|\theta)$ is the approximate posterior of the parameters.
> In practice, we approximate this expectation through sampling: $E_{q(W|\theta)}[P(Y|W,X)]\approx \frac{1}{N}\sum_{n=1}^{N} P(Y|W_n, X)$, where $W_n\sim q(W|\theta)$ is the n-th sample drawn from the approximate posterior, and $N$ is the total number of samples.
> When $N=0$, we use the mean of the weights to make predictions $P(Y|E_{q(W|\theta)}[W], X)$.
> Due to the *nonlinearity* of neural networks $P(Y|W,X)$, as $N \rightarrow \inf$, the average output converges to the expectation, but typically differs from the mean weight prediction. Thus,
> + the sample mean BLoB uses, $E_{q(W|\theta)}[P(Y|W,X)]\approx \frac{1}{N}\sum_{n=1}^{N} P(Y|W_n, X)$, is an unbiased estimator of $E_{q(W|\theta)}[P(Y|W,X)]$, but
> + BLoB is not an unbiased estimator of the mean prediction $P(Y|E_{q(W|\theta)}[W], X)$, and the mean prediction is not an unbiased estimator for the Bayesian prediction $E_{q(W|\theta)}[P(Y|W,X)]$ either. To summarize, when $N \rightarrow \infty$, $\frac{1}{N}\sum_{n=1}^{N} P(Y|W_n, X) \rightarrow E_{q(W|\theta)}[P(Y|W,X)] \neq P(Y|E_{q(W|\theta)}[W], X)$.
>
> To provide more context, we report the results for $N=0$ to $N=160$ on the WG-S dataset in the table below, demonstrating improved uncertainty estimation with increased number of samples but not converge to that of $N=0$.
>
> |                        | **N=0** | **N=1** | **N=2** | **N=3** | **N=4** | **N=5** | **N=10** | **N=20** | **N=40** | **N=80** | **N=160**  |
> | ---------------------- | ------- | ------- | ------- | ------- | ------- | ------- | -------- | -------- | -------- | -------- | ------ |
> | **ACC ($\uparrow$)**   | 71.44   | 65.20   | 66.28   | 66.91   | 67.63   | 68.20   | 68.31    | 68.04    | 68.15    | 68.31    | 68.15 |
> | **ECE ($\downarrow$)** | 19.71   | 19.72   | 14.60   | 12.82   | 11.27   | 10.58    | 9.51    | 9.47    | 9.25     | 8.75     | 8.75   |
> | **NLL ($\downarrow$)** | 0.84    | 0.8617  | 0.7355  | 0.6971  | 0.6766  | 0.6620  | 0.6395   | 0.6313   | 0.6233   | 0.6213   | 0.6215 |
>
>
> **Q2. Why is an ensemble used here? Shouldn’t the uncertainties come from the dropout variational posterior? Why are multiple models necessary?**
>
> We apologize for the confusion. By “ensemble of 10 LoRAs”, we meant to say “For MCD, we sample 10 times from the variational posterior distribution of LoRA with a dropout rate of $p = 0.1$ during inference." We will clarify this in the revision.
>
> **Q3. Will you open source the code on publication?**
>
> Thank you for your interest. The implementation of our method is straightforward and compatible with different LLMs. In fact, we have cleaned up the code, and will release the code upon acceptance of the paper.
>
> **Q4. Suggestions & Typos**
>
> We sincerely appreciate the reviewer's careful reading and for pointing out the typos. We will fix them in the revision as suggested.
>
> **Q5. Limitations**
>
> Please refer to **Q1. [Discussion of Limitations]** in **General Response**.

---

> > ### Comment · Reviewer_Lm4V · 2024-08-12
> > **Response to rebuttal**
> >
> > Thanks for the additional details, especially the derivation and experiment in Q1. I see that
> >
> > $$E_{q(W|\theta)}[P(Y | W, X)]  \neq P(Y | E_{q(W|\theta)} [W], X).$$
> >
> > Follow up: What's the intuition behind the mean prediction (RHS) achieving higher accuracy than the (approximate) Bayesian prediction (LHS), also for large $N$?
> >
> > Overall, the rebuttal partly addresses my concerns around novelty (Q2 and Q3 in "General response"), hence I increase my overall score by one point.

---

> > > ### Author Response · Authors · 2024-08-13
> > > **Thank You for Your Further Feedback**
> > >
> > > Thank you for your further feedback and for maintaining an open line of communication. We are glad that you found our response helpful and that it addressed your concerns. Below we address your follow-up comment on Q1.
> > >
> > > **[Higher Accuracy of Mean Prediction Compared to Bayesian Prediction]**
> > >
> > > This is a good question. By abandoning the modeling of the posterior distribution, mean prediction sacrifices some degree of calibration in exchange for improved accuracy. This empirical trade-off between accuracy and calibration has been noted in [3]. We will include this discussion in our revision.
> > >
> > > Lastly, we would like to express our gratitude once again for your insightful and constructive comments. They have significantly contributed to the improvement of our paper.
> > >
> > > [3] Stengel-Eskin, Elias, and Benjamin Van Durme. "Calibrated interpretation: Confidence estimation in semantic parsing." Transactions of the Association for Computational Linguistics 2023

---

### Author Rebuttal · Authors · 2024-08-05

# General Response
We thank all the reviewers for their valuable and constructive comments.
We are glad that they like our idea (Adn3) and find the problem we address ``"important"`` (Lm4V),
our paper ``"clear"``/``"well written"/"introduced in an adequate pace and order"`` (Lm4V, AGd6),
our method ``"nontrivial"``/``"necessary"``/``"promising"`` (AGd6, Adn3) as well as ``"justified"`` by theoretical and empirical arguments (AGd6),  and
our experiments showing that our method is ``"competitive"``/``"promising"``, of ``"efficiency"``/``"good performance"`` with ``"consistent gains over baseline approaches"`` (Adn3, Zor8, jN5u).

Due to space constraint (6000 characters), we cannot cover all questions, but we promise to address all questions and cite all related references in our revision. Below we address reviewers' common questions one by one.

**Q1. [Discussion of Limitations] (Lm4V, Zor8, Adn3, AGd6)**

This is a good suggestion. We will add a Limitations section in the revision, which is summarized below:
+ Currently, BLoB is not suitable for training-free tasks (direct operation on LLMs during inference), which are interesting future work.
+ As the common disadvantage of VI algorithms, BLoB also requires sampling $N$ times during inference.
+ With a limited number of samples, the presence of unavoidable noisy samples can affect the algorithm's performance. Sampling a computationally acceptable number of times (e.g., $N=10$) can mitigate this issue.

**Q2. [Reproducing LAP's Results on the ARC-C Dataset] (Lm4V, Adn3)**

Our experimental setup strictly adheres to the original LoRA [2] framework, which has the **same hyperparameters** for **all datasets and methods** (learning rate, fine-tuning iterations, etc.). This widely adopted approach in LoRA-based research enhances our work's relevance and potential impact.

Yang et al. [1] (LAP) employ a different setup in their original paper, incorporating early stopping to ensure a reasonable MAP for subsequent LA. However, this often results in reduced accuracy (over 1% deficit), prompting our decision to diverge from their setting.

Nonetheless, we have diligently reproduced Yang et al. [1] within our framework, utilizing exactly their released code, with exactly the same hyperparameter configurations. However, we were unable to find a set of hyperparameters that consistently works for LAP across all datasets, leading to its poor performance on the ARC-C dataset.

To achieve competitive performance with LAP on the ARC-C dataset, we deviated from the unified setting (i.e., the same hyperparameter configuration for all datasets), allowing LAP to have different hyperparameter configurations for different datasets. Note that this creates an unfair advantage for LAP.

Specifically, we conducted an exhaustive grid search on 3 hyperparameters, resulting in the optimal configuration below:
- Dropout rate: 0.1
- Learning rate: 5e-5
- Early stopping: At 5000th iteration (out of 10000)

The table below shows the corresponding results (along with the original BLoB results for reference):

| Metrics                | LAP          | BLoB ($N=10$) |
| ---------------------- | ------------ | ------------- |
| **ACC**   | 66.78 (0.69) | **68.81 (1.09)** |
| **ECE** | 16.25 (2.61) | **9.59 (1.88)**  |
| **NLL** | 1.03 (0.04)  | **0.78 (0.02)**  |

Note that BLoB (under the unified setting) still outperforms LAP even when LAP has the unfair advantage of allowing different hyperparameter configurations for different datasets. This showcases our BLoB robust performance improvement.

During our reimplementation, we observed that LAP is highly dependent on and sensitive to MAP estimation. It frequently fails, corroborating your previous observation about LAP's potential disadvantage: sub-optimal MAP convergence significantly impacts LAP's performance.

We will include the discussion above in the revision as suggested.

[1] Yang, Adam X., et al. "Bayesian Low-rank Adaptation for Large Language Models." ICLR 2024.

**Q3. [Novelty of BLoB] (Lm4V)**

While using Bayesian methods to address inaccurate uncertainty estimation in neural networks is not new, it is a **nontrivial** contribution to apply these techniques effectively to Large Language Models (LLMs) and demonstrate their practicality.

Laplace Approximation (**LA**) and Variational Inference (**VI**) are **two major approximate Bayesian inference paradigms**. While Yang et al. [1] validated **LA**'s effectiveness for LLMs, **VI** remains unexplored in this context. Our BLoB, as **the first representative VI approach on LLMs**, addresses this major research gap by demonstrating performance comparable or superior to LAP. As Reviewer AGd6 noted, while ``"the proposed algorithm seems like a straightforward combination of LoRA and BBB, the paper mentions that certain modifications are necessary for the algorithm to work in practice. These modifications are a core part of the paper's contribution, and they are discussed thoroughly and justified with a mix of theoretical and empirical arguments."``

To summarize our BLoB's novel contributions:
+ BLoB is, to our knowledge, the first VI method for LLM fine-tuning; it demonstrates effective uncertainty estimation across diverse datasets.
+ We provide theoretical analysis, proving the feasibility of optimizing the full-weight variational distribution in the low-rank space of weight update matrices, supported by empirical evidence of its effectiveness and efficiency.
+ To address BBB's consistent failures, our BLoB introduces:
  + a novel approximate posterior parameterization method, enabling fast convergence and accurate uncertainty estimation within limited training iterations, and
  + a novel KL re-weighting scheme that effectively balances data likelihood and model complexity during training.

We will include the discussion above in the revision as suggested.

---

### Decision · Program_Chairs · 2024-09-25

**Decision:**

Accept (poster)

**Comment:**

The paper proposes an algorithm to learn a (low-rank) variational posterior distribution over (a subset of) LLM weights during fine-tuning by combining Low-Rank Adaptation (LoRA) and Bayes by Backprop (BBB). To make the algorithm work in practice, several nontrivial modifications are introduced, such as certain parameterisations and optimisation strategies. Finally, the proposed method is empirically evaluated on a variety of in-distribution and out-of-distribution benchmarks and compared to relevant baselines.

All reviewers provided accept rating after the author effective rebuttal, but one is “borderline” and two are “weak”.
The most quoted strength of this work is the thorough experimental comparison that shows strong results compared to other LORA-based fine-tuning approach. The approach is also simpler that other Bayesian methods.

The most quoted weakness is about the contribution.
- Reviewer Adn3: “Blob is  mean-field Variational inference on LoRA parameters”
- Reviewer Lm4v:  “Novelty is limited compared to Yang et al. [1], which implements a very similar idea via Laplace approximations instead of the Bayes-by-Backprop variational posterior.”

But the author detailed rebuttal show that their method is a substantial progress over the Laplace LoRA, being much more stable and practical. This is confirmed by reviewer AGd6:
“While the proposed algorithm seems like a straightforward combination of LoRA and BBB, the paper mentions that certain modifications are necessary for the algorithm to work in practice. These modifications are a core part of the paper's contribution, and they are discussed thoroughly and justified with a mix of theoretical and empirical arguments.”

Reviewers strongly disagree about the presentation, with ratings ranging from 1 (poor) to 4 (excellent). The AC, who is not an expert in LoRA or Bayesian methods, found the paper easy to read and understand.